

# Forest tree species distribution for Europe 2000–2020: mapping potential and realized distributions using spatiotemporal machine learning

Carmelo Bonannella[1,2], Tomislav Hengl[2], Johannes Heisig[3], Leandro Parente[2], Marvin N. Wright[4,5], Martin Herold[1,6] and Sytze de Bruin[1]

[1] Laboratory of Geo-Information Science and Remote Sensing, Wageningen University and Research, Wageningen, The Netherlands
[2] OpenGeoHub, Wageningen, The Netherlands
[3] Institute for Geoinformatics, University of Münster, Münster, Germany
[4] Leibniz Institute for Prevention Research and Epidemiology – BIPS, Bremen, Germany
[5] University of Bremen, Bremen, Germany
[6] Section 1.4 Remote Sensing and Geoinformatics, GFZ German Research Centre for Geosciences, Potsdam, Germany

Corresponding author
Carmelo Bonannella,
carmelo.bonannella@opengeohub.org

## ABSTRACT

This article describes a data-driven framework based on spatiotemporal machine learning to produce distribution maps for 16 tree species (*Abies alba* Mill., *Castanea sativa* Mill., *Corylus avellana* L., *Fagus sylvatica* L., *Olea europaea* L., *Picea abies* L. H. Karst., *Pinus halepensis* Mill., *Pinus nigra* J. F. Arnold, *Pinus pinea* L., *Pinus sylvestris* L., *Prunus avium* L., *Quercus cerris* L., *Quercus ilex* L., *Quercus robur* L., *Quercus suber* L. and *Salix caprea* L.) at high spatial resolution (30 m). Tree occurrence data for a total of three million of points was used to train different algorithms: random forest, gradient-boosted trees, generalized linear models, k-nearest neighbors, CART and an artificial neural network. A stack of 305 coarse and high resolution covariates representing spectral reflectance, different biophysical conditions and biotic competition was used as predictors for realized distributions, while potential distribution was modelled with environmental predictors only. Logloss and computing time were used to select the three best algorithms to tune and train an ensemble model based on stacking with a logistic regressor as a meta-learner. An ensemble model was trained for each species: probability and model uncertainty maps of realized distribution were produced for each species using a time window of 4 years for a total of six distribution maps per species, while for potential distributions only one map per species was produced. Results of spatial cross validation show that the ensemble model consistently outperformed or performed as good as the best individual model in both potential and realized distribution tasks, with potential distribution models achieving higher predictive performances (TSS = 0.898, $R^2_{logloss}$ = 0.857) than realized distribution ones on average (TSS = 0.874, $R^2_{logloss}$ = 0.839). Ensemble models for *Q. suber* achieved the best performances in both potential (TSS = 0.968, $R^2_{logloss}$ = 0.952) and realized (TSS = 0.959, $R^2_{logloss}$ = 0.949) distribution, while *P. sylvestris* (TSS = 0.731, 0.785, $R^2_{logloss}$ = 0.585, 0.670, respectively, for potential and realized distribution) and

*P. nigra* (TSS = 0.658, 0.686, $R^2_{logloss}$ = 0.623, 0.664) achieved the worst. Importance of predictor variables differed across species and models, with the green band for summer and the Normalized Difference Vegetation Index (NDVI) for fall for realized distribution and the diffuse irradiation and precipitation of the driest quarter (BIO17) being the most frequent and important for potential distribution.

On average, fine-resolution models outperformed coarse resolution models (250 m) for realized distribution (TSS = +6.5%, $R^2_{logloss}$ = +7.5%). The framework shows how combining continuous and consistent Earth Observation time series data with state of the art machine learning can be used to derive dynamic distribution maps. The produced predictions can be used to quantify temporal trends of potential forest degradation and species composition change.

# INTRODUCTION

Reforestation and forest restoration are considered key strategies for tackling climate change by enhancing $CO_2$ sequestration (*Lefebvre et al., 2021*; *Domke et al., 2020*; *Nave et al., 2019*). Under the European Green Deal and the European biodiversity strategy for 2030, the European Union has committed to plant at least three billion additional trees by 2030 (*European Commission, 2021*). At the same time, tree deaths due to bark beetle infestations and increased drought fueled by a warming climate have reduced the total forest area of Germany by 2.5% since 2018 (*Popkin, 2021*). Obtaining reliable information on forest tree species distribution in both space and time is now urgently required for stakeholders and decision-makers in order to develop effective forest management and adaptation strategies (*Keenan, 2015*).

Understanding the range, constraints and drivers of species distribution has always been a primary goal of ecology (*Andrewartha & Birch, 1954*). However, only with the advent of Geographical Information Systems (GIS) and the usage of extensive digital maps of environmental variables were ecologists able to access powerful enough tools to study species distributions at landscape scales (*Franklin, 1995*). Progress in this direction has given rise to a new field called Species Distribution Modelling (SDM) (*Franklin, 2010*): maps of species ecological niches are made by associating values of different predictors to known locations of the target species and then used to predict distribution in geographic space where no field data for the target species is available. Commonly, SDMs rely on climatic or bioclimatic factors at a coarse spatial resolution (≥1 km) while in the temporal dimension long time averages (30–50 years) are often used (*Iturbide, Bedia & Gutiérrez, 2018a*). Whilst forest distribution maps are often used to guide management decisions happening at local scales, the potential impact of differences in resolution of the predictor variables on the results is often overlooked (*Porfirio et al., 2014*). For conservation

purposes, previous studies have shown how distribution maps with high spatial resolution (<100 m) and slightly lower prediction accuracy are actually more useful than coarser (>250 m) but more accurate maps (*Manzoor, Griffiths & Lukac, 2018*; *Guisan et al., 2013*; *Gottschalk et al., 2011*; *Prates-Clark, Saatchi & Agosti, 2008*). Therefore, even at the cost of overall map accuracy, finer spatial resolution maps are more valuable for practical use. For these reasons, Earth Observation (EO) data, and specifically the use of high spatial resolution data, have grown in use for SDM applications (*Gelfand & Shirota, 2021*; *Pérez Chaves, Ruokolainen & Tuomisto, 2018*; *Hefley & Hooten, 2016*).

In addition to the clear need for finer spatial resolution mapping, there are similar needs to drive research towards producing finer temporal resolution mapping. This is due to the recent and relatively swift change in disturbance regimes and weather patterns, which are significantly altering the ecological niches of tree species on a temporal scale of less than a few decades instead of centuries. The impact of these changes on tree species has become more noticeable from year to year, with growth decline (*Martinez del Castillo et al., 2022*) and increased mortality rates (*Senf, Sebald & Seidl, 2021*; *Senf et al., 2018*) demonstrated in literature as already occurring across large forested areas. Including the temporal domain in tree species distribution studies is therefore fundamental to capture the temporal evolution of these change processes. However no general consensus has yet been reached on the influence of these new high spatial and temporal resolution data sources on SDM performances. The inclusion of spatiotemporal data sources in SDM studies requires taking an additional effort when choosing the appropriate modeling technique, a task that has proved to be difficult even with traditional spatial-only data sources (*Elith & Graham, 2009*), let alone when also attempting to include the temporal dimension.

Aside from spatial and temporal considerations of predictor variables and species observations, in the last decade ecologists have conducted hundreds of studies purely to determine which modeling methods best suit the needs of SDM. Model choices have thus far proven to be highly impactful, with distribution maps derived with different models from the same dataset leading to quite opposite conclusions (*Araújo & New, 2007*; *Pearson et al., 2006*). Inter-model variability in projections has been tackled using ensemble modeling, where numerous independent models are fit using a range of methods applied to the same input data while the outputs of the individual models are aggregated into the final prediction. Ensemble modeling is a solution to high model variance and it has been demonstrated that reducing variance also reduces the effect of model overfitting and extrapolation (*Zhou, 2019*). This is achieved at the cost of increased model complexity, reduced model interpretability, and increased computational time (*Zhou, 2019*). As such, the few examples of ensemble modeling approaches that have been investigated for SDM applications are limited to *mean*, *median* and *weighted average* approaches (*Hao et al., 2019*). These approaches are intuitively simple to implement and interpret, and involve, in the first two cases, just taking the mean or median of the predictions of the individual models as the final prediction. The weighted average approach is similar but scales the predictions by weights assigned based on predictive performances of the models obtained from cross validation. A robust ensemble technique that, to our knowledge, has not been

tested yet for SDM is *stacking* or *stacked generalization*. In this approach outputs made by the individual models are the inputs of a *meta-learner* (*i.e.*, a model that learns from other models) which then produces the final prediction (*Wolpert, 1992*).

We tested this ensemble technique on European forest tree species distribution. There is no shortage of information on European tree species distribution: the European Atlas of Forest Tree Species is among one of the largest data sources with information on forest tree species for Europe (*San-Miguel-Ayanz et al., 2016*). It describes in detail the autoecology of 76 different forest tree species and provides geographical information on each species in the form of chorological maps, probability of presence maps and maximum habitat suitability maps. Recently, the Atlas has been further expanded with future projections in different climatic scenarios (*Mauri et al., 2022*). While these predictions are certainly useful to determine potential species composition of European forests, new methods are now needed to deal with the more and more attention to reproducibility of studies (*Fidler et al., 2017*), increasing spatial and temporal resolution of predictor variables (*Zhu et al., 2019*) and availability of ecological *"big data"* (*i.e.*, gathered by multiple sources such as sensors, cameras etc.) (*Hampton et al., 2013*). Furthermore, SDM studies use high-dimensional data which is often non-linear and does not meet assumptions of conventional statistical procedures (*Zhang & Li, 2017*). For this reason, and thanks to the exponential increase in computing power of the last decade (*Gorelick et al., 2017*), solutions such as machine learning (ML) algorithms have recently become very popular for SDM studies. ML tries to learn the relationship between the response and the predictors through the observation of dominant patterns (*Breiman, 2001b*). Contrary to traditional statistical models, no kind of ecological assumptions are explicitly embedded in ML algorithms: ML can be especially useful when dealing with data gathered without a specific and rigorous sampling design (*Bzdok, Altman & Krzywinski, 2018*). ML algorithms have great potential to analyze the large amount of data available nowadays, enabling the mapping and monitoring of changes on multiple geographical scales in a timely manner through reproducible research (*Gobeyn et al., 2019*).

In this sense, the objectives of this study were (a) to test different ML algorithms to develop a framework for modeling species distribution in space-time, (b) to assess the importance of various sources of EO data on model performances for mapping tree species distributions and (**c**) to explore and quantify the specific importance of high resolution data on model performances.

## MATERIALS AND METHODS

### General workflow

We modeled potential and realized distribution for 16 forest tree species for continental Europe for the time period January 2000–December 2020 using a spatio-temporal ML approach. The general workflow used to derive the distribution maps is shown in Fig. 1. We modeled the potential distribution as a baseline to assess the importance of EO data sources: we used only environmental predictors (*i.e.*, temperature, precipitation, wind speed, water vapor and topographical variables) and environmental absences (*i.e.*, location with known environmental conditions not suitable for the target species, following the

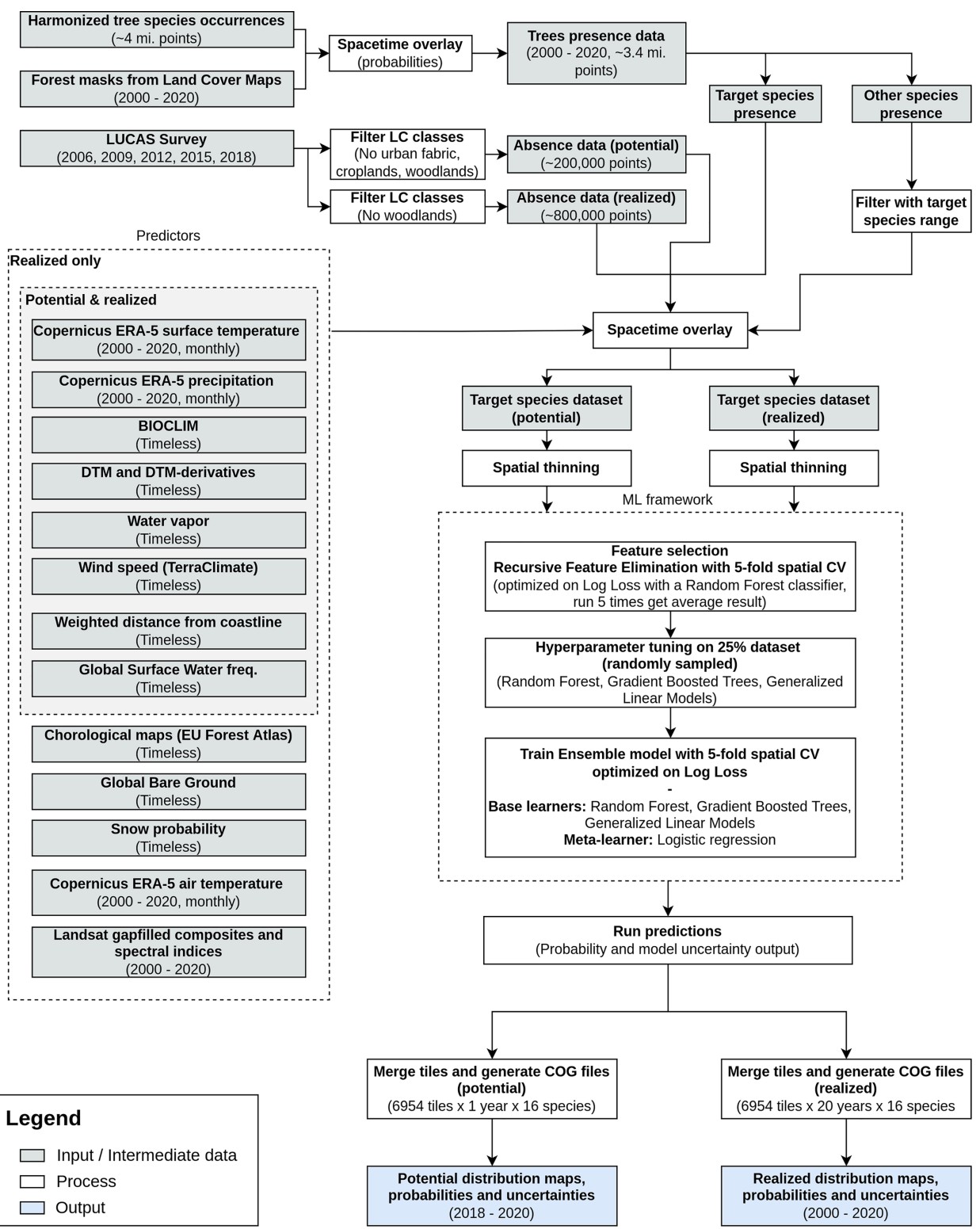

**Figure 1 General workflow illustrating the preparation of the point data, the predictor variables used, model building (feature selection—hyperparameter optimization—training) and preparation of distribution maps for one species.** The process was identically replicated for all the species.

definition used by *Lobo, Jiménez-Valverde & Hortal (2010)*) to produce a neutral model for baseline species potential.

As an additional source of homogeneously distributed true absence data we used the Land Use/Cover Area Survey (LUCAS) (*EUROSTAT, 2017*) dataset: *in-situ* observations of land use and land cover distributed on a 2 × 2 km grid covering the whole European Union (see *d'Andrimont et al. (2021)* for more information and https://ec.europa.eu/eurostat/web/lucas/data/lucas-grid for the official grid). Final prediction maps show the probability of presence (0–100%) of at least one individual of the target species in the area covered by a 30 m pixel. Probability of presence is relative to the mapped target species, irrespective of the potential co-occurrence of other species in the same 30 m pixel and should not be confused with the absolute abundance or proportion of each species in the pixel area. The sum of the presence probabilities of different species in the same pixel can thus exceed 100%. We produced one potential distribution map and six realized distribution maps for each species: the assumption is that the conditions in the study area that determine the potential distribution of the species did not change over the time period analyzed; this does not hold for the realized distribution. We split the time period analyzed in six time windows according to the following scheme: (1) 2000–2002, (2) 2002–2006, (3) 2006–2010, (4) 2010–2014, (5) 2014–2018 and (6) 2018–2020.

To ensure transparent reporting and reproducibility, we described the dataset according to the ODMAP protocol suggested by *Zurell et al. (2020)*. We implemented the workflow in the Python (*Van Rossum & Drake, 2009*) and R (*R Core Team, 2021*) programming languages. More technical details on preprocessing steps and packages used according to ODMAP (*Zurell et al., 2020*) are presented in Table S1 (*Bonannella et al., 2022*).

## Study area

The study area covers the European continent, that is all countries included in the Corine Land Cover (CLC) database (*Büttner et al., 1998*) except Turkey (Fig. 2). European forests cover 33% of the continent's land area. Owing to the variety of climatic conditions across both latitudinal and longitudinal gradients, twelve out of the 20 FAO Forest Ecological Zones are represented in European forests (*de Rigo et al., 2016*). The European Atlas of Forest Tree Species (*San-Miguel-Ayanz et al., 2016*) reports detailed information for a total of 76 forest tree species. From those, a selection of 16 were chosen and modelled in this study. The complete list of species is presented in Table S1.

## Training points

### Preparing and combining legacy occurrence points

A total of 2,454,997 tree species occurrence points from three different sources were gathered. The majority of points (71%) comes from the Global Biodiversity Information Facility (GBIF). National Forest Inventory (NFI) data from multiple EU member states published by *Mauri, Strona & San-Miguel-Ayanz (2017)* forms another 23% of the dataset. The remaining 6% comes from the LUCAS dataset.

Entries were filtered for species included in the European Atlas of Forest Tree Species (*San-Miguel-Ayanz et al., 2016*). Occurrences with a taxonomy rank other than species or

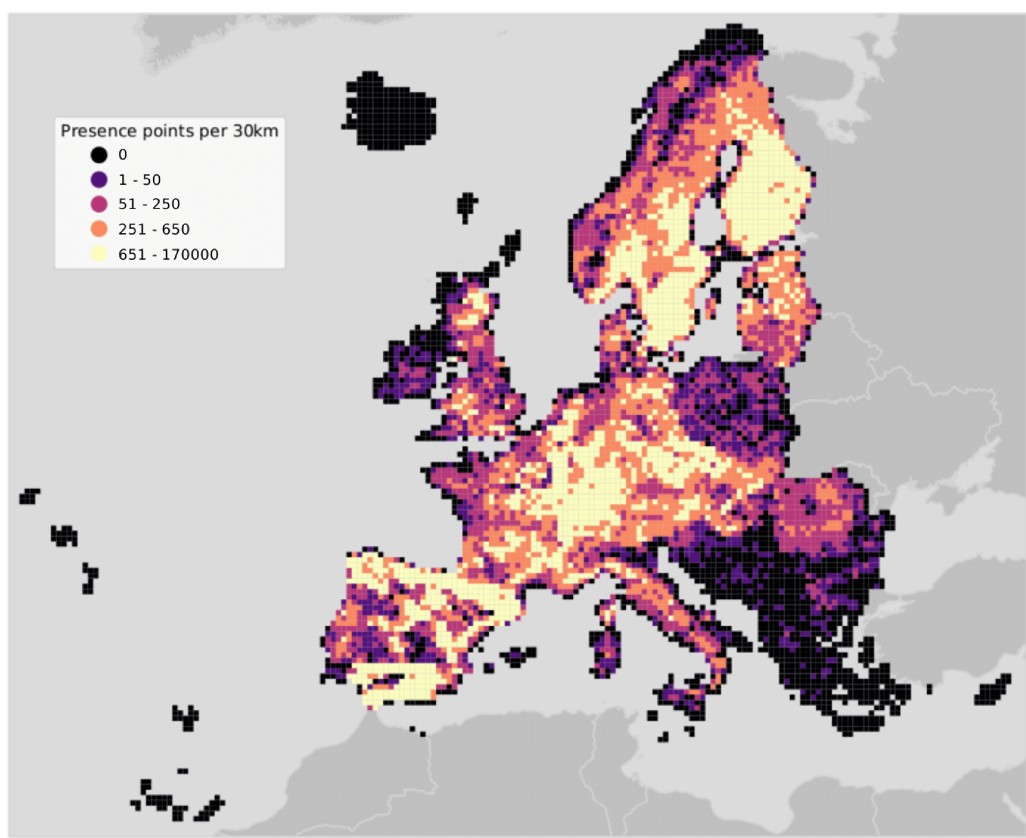

**Figure 2 Map of the study area showing presence points only.** Points are aggregated at a coarse resolution (30 km) scale and absence points are omitted for visualization purposes.

genus were omitted. Same applies to points which had flags indicating serious location issues (*i.e.*, missing coordinates). Geometries were re-projected to coordinate reference system ETRS89/LAEA Europe (EPSG: 3035). A high resolution land mask for Europe (*Hengl et al., 2020*) was applied to further exclude misplaced occurrence points. GBIF taxon and genus keys were derived for the other two data sources. Quality flag variables for location accuracy and date were established from existing metadata to indicate potentially problematic entries. The harmonized point dataset has information on species and genus (including respective GBIF keys), year of observation, country, original data source, citation, and license among other auxiliary variables. The dataset was published separately (*Heisig & Hengl, 2020*).

We used yearly forest masks derived from *Witjes et al. (2022)* to decide upon including point data lacking the year of observation. *Witjes et al. (2022)* provides yearly probability maps at 30 m for the 2000–2019 for 43 land cover classes according to the CLC level three legend. We overlaid the points with the probability maps with prevalent forest (classes: 311, 312, 313 and 323) or woodland-shrub (324, 333) cover. Points were used only if the probability value extracted for at least one of the classes was ≥50% for all the years considered. Each unique combination of longitude, latitude and year was then considered as an independent sample. An additional quality flag was added to distinguish points

coming from this operation and the points with original year of observation coming from source datasets.

### Preparing non-occurrence points

A total of 883,630 land cover points were gathered from the LUCAS database as provided by Eurostat and used as absence data. All LUCAS survey data (2006, 2009, 2012, 2015 and 2019) was used: each survey was first downloaded individually and then aggregated. As for the occurrence points, spatial and temporal information were used to uniquely identify one observation. All main land cover classes were used for selecting observations for the absence dataset with the exception of class C (Woodland class), as points belonging to that class already served the selection of presence data. For potential distribution only, points coming from human influenced land cover classes (class A and B) were also excluded. This choice was taken assuming that cities (class A) and croplands (class B) could be suitable areas for the target species if only environmental criteria are met. Two presence-absence datasets were produced for each species, one to be used for potential distribution and one for realized distribution. Locations in space and time of the target species were considered to be true presences, while presence locations of other species and observations from the LUCAS dataset were assumed to be the true absence locations. Presence locations of other species used as true absence were additionally filtered by overlaying them with a rasterized chorological map downloaded from the European Atlas of Forest Tree Species portal for each of the target species. Only points falling outside the geographical extent of the target species chorological map were used as absence locations for modeling.

### Spatial thinning

Combining different data sources to generate the tree occurrence points produced a dataset with unknown sampling design, while LUCAS points are regularly distributed across the whole study area. To overcome the problem of uneven sampling intensity and spatial clustering, we applied a spatial thinning procedure using the *spThin* R package (*Aiello-Lammens et al., 2015*). A distance of 2 km was considered as minimum distance between the points, to harmonize the sampling intensity between presence and absence data. The procedure was repeated 10 times: at each iteration, the algorithm randomly removes one observation from the dataset until no observation is left with a nearest neighbor closer than the thinning distance. Among the 10 datasets obtained, the one with the largest number of records was retained and used for modeling. However, the package was not developed for large datasets: the implementation of the thinning algorithm cannot be processed in parallel and computation time can take even days with a number of observations ≥3,000. Due to these computational constraints, we first overlaid the points with a $10 \times 10$ km grid and ran the thinning procedure per tile. Results of this operation are shown in Table S2 and Fig. S1 (*Bonannella et al., 2022*).

## Predictor variables

A total of 305 harmonized variables covering continental Europe at different spatial resolution were used as predictors to model the realized distribution of the species. In this

**Table 1 Table with Landsat-derived spectral indices used in this study.**

| Spectral index | Abbreviation | Formula | Reference |
|---|---|---|---|
| Enhanced vegetation index | EVI | $2.5 \times \dfrac{NIR - RED}{NIR + 6 \times RED - 7.5 \times BLUE + 1}$ | *Huete et al. (2002)* |
| Enhanced vegetation index 2 | EVI2 | $2.5 \times \dfrac{NIR - RED}{NIR + 2.4 \times RED + 1}$ | *Jiang et al. (2008)* |
| Modified soild adjusted vegetation index | MSAVI | $\dfrac{(2 \times NIR + 1) - \sqrt{(2 \times NIR + 1)^2 - 8 \times (NIR - RED)}}{2}$ | *Qi et al. (1994)* |
| Normalized burned ratio | NBR | $\dfrac{NIR - SWIR2}{NIR + SWIR2}$ | *Key & Benson (1999)* |
| Normalized difference vegetation index | NDVI | $\dfrac{NIR - RED}{NIR + RED}$ | *Tucker (1979)* |
| Normalized difference wetness index | NDWI | $\dfrac{NIR - SWIR1}{NIR + SWIR1}$ | *Gao (1996)* |
| Soil Adjusted Vegetation Index | SAVI | $(1 + 0.5) \times \dfrac{NIR - RED}{(NIR + RED + 0.5)}$ | *Huete (1988)* |

study we included both dynamic (*i.e.*, time-series of data of different temporal resolution) variables covering the time period January 2000–December 2020 and static (*i.e.*, variables not expected to change during the modelled time period) variables. A subset of only 103 variables was used instead to model the potential distribution (see Fig. 1). All data was reprojected in the coordinate reference system ETRS89/LAEA Europe (EPSG: 3035) before the analysis.

### Dynamic data

We used a reprocessed version of Landsat ARD data provided by Global Land Analysis and Discovery (GLAD) (*Potapov et al., 2020*): time series used in this study covers the period 1999–2020. Cloud and cloud shadow pixels were removed from the images, maintaining only the quality assessment-QA values labeled as clear-sky. Afterwards, individual images were averaged by season according to three different quantiles (25th, 50th and 75th) and the following calendar dates for all periods:

- Winter: December 2 of previous year until March 20 of current year,
- Spring: March 21 until June 24 of current year,
- Summer: June 25 until September 12 of current year,
- Fall: September 13 until December 1 of current year.

84 images (3 quantiles × 4 seasons × 7 Landsat bands) were produced for each year. Missing values were imputed using the *Temporal Moving Window Median* algorithm. For more details on the preprocessing of Landsat data for this study see *Witjes et al. (2022)*. Seven different spectral indices (see Table 1) were computed for each year and season using the 50th quantile only, for a total of 7 × 4 = 28 spectral indices variables per year.

A reprocessing of the ERA5 Land hourly dataset has been used to have monthly aggregates of air temperature (2 m above ground), surface temperature and precipitation. Original ERA5 data was aggregated to daily data, and subsequently to monthly data, with

increased resolution (1 km) using CHELSA data (*Karger et al., 2020*): in this way the general spatial and temporal pattern of ERA5 Land dataset was kept while using the fine spatial detail coming from the CHELSA dataset. For air and surface temperature we obtained the monthly minimum, mean and maximum, while for precipitation the monthly sum for a total of 84 climatic time series layers.

### Static covariate datasets

As additional static covariates, we used the following datasets:

- 19 bioclimatic variables (*Hijmans et al., 2005*) for the period 1979–2013 to provide a baseline of the actual state of the climate; we used bioclimatic variables from the CHELSA dataset since it has been claimed to better match data from meteorological stations than WorldClim (*Karger et al., 2017*),
- 50 chorological maps downloaded from the European Atlas of Forest Tree Species web portal. Chorological maps provide a qualitative overview of the spatial distribution of a species over an area, differentiating between native and introduced. We considered both the native and introduced areas as the potential distribution of a species for the time period covered by the study,
- Global bare ground cover from *Hansen et al. (2013)*. The layer provides information on bare ground cover on a percent (1–100) scale,
- Solar direct and diffuse irradiation,
- 13 cloud fraction layers (monthly averages and annual average) derived from MODIS (*Wilson & Jetz, 2016*),
- Digital terrain model (DTM) for Europe (*Hengl et al., 2020*) and DTM-derived (slope, hillshade) variables,
- Easterness, northness (*Olaya, 2009*), and positive and negative openness,
- Probability of surface water occurrence at 30 m resolution derived from Landsat time series (*Pekel et al., 2016*),
- Height above nearest drainage (HAND) and flow accumulation area at 90 m resolution from the MERIT Hydro global hydrography datasets,
- Long-term flood hazard map calculated on a 500 years time period (*Dottori et al., 2016*),
- Water vapor pressure (kPa) based on the WorldClim2.1 dataset (*Fick & Hijmans, 2017*),
- Long-term snow probability based on MODIS (MOD10A2) and available at https://doi.org/10.5281/zenodo.5574953,
- Monthly wind speed (1998–2018) from TerraClimate.

For more details on spatial and temporal resolution, preprocessing and data sources see the Supplemental Material (*Bonannella et al., 2022*).

## Feature selection

Features for potential and realized distribution for each species were selected using the Recursive Feature Eliminitation (RFE) strategy, implemented in the scikit-learn library (*Pedregosa et al., 2011*). For each combination of species and modelled distribution we

trained a random forest classifier (num.trees = 50, default values were used for the other parameters): RFE fits the model and removes the weakest feature until a specified number of features is reached, then ranks the importance of the features based on the model's coefficients (for regression-based models) or feature importance (for random forest).

The minimum number of features was not known before hand: to select this number, we ran the Recursive Feature Elimination with a spatial five–fold Cross Validation (RFECV), using the logarithmic loss, or logloss, as a scoring estimator. Logloss is one of the most robust performance metrics when it comes to imbalanced datasets (*Ferri, Hernández-Orallo & Modroiu, 2009*). Logloss is indicative of how close the predicted probability for an observation *i* is to the corresponding label *y*. For binary classification with label $y \in 0, 1$ the overall logloss was calculated as:

$$Logloss = -\frac{1}{N}\sum_{i=1}^{N} y_i \cdot \ln[p(y_i)] + (1 - y_i) \cdot \ln[1 - p(y_i)] \tag{1}$$

where $N$ is the total number of observations and $p(y_i)$ is the predicted probability for an observation with $y_i = 1$. It follows that values close to 0 indicate high prediction performances, with logloss = 0 being a perfect match, and values that are positive to infinite are progressively worse scores. For comparison, the value of logloss for random assignment depends on the number of classes (a) and the prevalence of the classes (b): for binary classification and a balanced (50:50) dataset with $N = 10$ observations, the Eq. (1) gives a value of 0.69.

We ran the RFECV on a 25% random subsample for each species and modelled distribution; this operation was replicated five times. For each iteration we selected the minimum of the logloss function (Fig. S2) and the averaged result was then used as the minimum number of features for the RFE.

## Model building and evaluation

### Modeling methods

To build an ensemble model, we decided to compare predictive performances and computing time (hyperparameter tuning—cross validation—prediction time) of different machine learning algorithms on a random 25% subset of observations for both potential and realized distribution datasets. A detailed workflow of this process is shown in Fig. S3. We decided to conduct this test on seven different species: choice of the species was based on the spatial distribution of the training points and the ratio between presence and absence points. In this way, algorithms performances could be tested on different ecological conditions (latitudinal and longitudinal gradient) and imbalance of classes. The species selected were: *A. alba*, *C. sativa*, *F. sylvatica*, *P. abies*, *P. halepensis* and *P. sylvestris*.

We compared seven different algorithms: Random Forests (RF) (*Breiman, 2001a*), Gradient-boosted trees (GBT) (*Friedman, 2002*), Classification trees (CART) (*Therneau & Atkinson, 1997*), Generalized Linear Models (*Nelder & Wedderburn, 1972*) with Lasso regularization (*Tibshirani, 1996*) (just GLM from now on), C5.0 (*Quinlan, 1986*), K-nearest neighbor (KNN) (*Fix & Hodges, 1989*) and Artificial Neural Network (ANN)

(*Ripley & Venables, 2017*). Analyses were conducted using the *mlr* package (*Bischl et al., 2016*). For each algorithm, a hyperparameter space was defined: combinations of hyperparameters were generated per model based on a grid search of five steps per hyperparameter. More details on the hyperparameter space are available in Table S3.

### Selecting component models

We evaluated each combination of hyperparameters by comparing logarithmic loss values during a five–fold spatial cross validation replicated 5 times: we used spatial instead of normal cross validation for hyperparameter tuning because it reduces overoptimistic performance results in the presence of strong data clustering (*Schratz et al., 2019*). We used the tile ID produced in the tiling system for Europe as the blocking parameter in the training function in *mlr*. All the compared algorithms were used in "*probability*" mode, that is, predicting for each observation in the dataset a probability value for presence (class 1) and absence (class 0). Besides the performance achieved in the logloss metric, computing time for the hyperparameter tuning, a five–fold spatial cross validation and prediction time for a 30 km tile were also considered as additional criteria: we calculated the computing time only for the species that had the highest computational costs (*P. sylvestris*). This gave us an estimate of how long the process of training each component model could take during the building of the ensemble model. We used logloss performances as the first criteria to choose the component models: only in the case of two or more methods performing within one standard deviation from the average performance, we chose the computationally fastest.

### Training ensemble model using stacking

Stacked generalization involves combining predictions made by level 0 models and using them as training data for a level 1 model (or meta-learner from now on) (*Wolpert, 1992*). To limit overfitting in the training data, we used a five–fold spatial cross validation: the out-of-fold predictions were used to build a level 1 training dataset for the meta-learner. We used logistic regression as a meta-learner, which is usually the most used model for classification problems (*Gomes et al., 2012*). Final predictions are delivered as probability maps (0–100%) for presence together with model uncertainty maps: we consider as model uncertainty the standard deviation of the predicted values of the base learners.
The principle is that the higher the standard deviation the more uncertain the model is regarding the right value to assign to the pixel (*Brown, Bhuiyan & Talbert, 2020*).

### Variable importance assessment

To assess to what extent the three level 0 models used different parts of the available feature space and the agreement between these models, we compared the variable importance when possible. For RF and CART we used Gini importance, for C5.0 the "*percentage of training set samples that fall into all the terminal nodes after the split*" (*Quinlan, 1986*), for GBT the gain metric (*Shi et al., 2019*) and for GLM the coefficients for the minimum fitted
value of $\lambda$ (*Hastie, Qian & Tay, 2016*). To better analyze the results, we aggregated the whole set of variables in seven macro-classes:

- Climate (*i.e.*, precipitation, wind speed, water vapor, snow probability etc.),
- Temperature (*i.e.*, time series of recorded temperatures for the observed time period),
- Bioclim (*i.e.*, bioclimatic variables from CHELSA),
- Topography (*i.e.*, DTM and DTM-derivative variables),
- Landsat Band (*i.e.*, all percentiles, all seasons),
- Distribution (*i.e.*, species distribution maps from European Atlas of Forest Tree Species),
- Spectral Index (*i.e.*, spectral indices derived from Landsat bands).

### Model evaluation

Predictive performance of the ensemble model was assessed through spatial five–fold cross-validation repeated 5 times (*Roberts et al., 2017*) with logloss as performance metric. To investigate if the ensemble model outperformed the component models, we compared results of the spatial cross validation of the ensemble with the results of the component models. To be able to compare performances between different species, we converted logloss performances using the following formula:

$$R^2_{\text{logloss}} = 1 - \frac{Logloss_{\text{m}}}{Logloss_{\text{r}}} \tag{2}$$

where $Logloss_{\text{m}}$ is the performance achieved by the model and $Logloss_{\text{r}}$ is the value for random logloss, used as a baseline for predictive performances. Values close to 1 indicate high predictive performances, while values close to 0 indicate lower performances, with 0 meaning that the model is no better than a random guess. We also reported a threshold-dependent metric, the True Skill Statistic (TSS) and a threshold independent metric, the area under ROC curve (AUC), as they are commonly used metric to evaluate SDMs predictive performances (*Chakraborty et al., 2021*; *Shabani, Kumar & Ahmadi, 2018*). TSS was computed using the default threshold value (0.5) when assigning predicted probabilities values to the presence or absence class. Logloss is one of the least sensitive metric to prevalence (*Ferri, Hernández-Orallo & Modroiu, 2009*), hence our choice of logloss as a primary performance metric to compare different models coming from different training datasets.

To assess the effect of high resolution products on predictive performances, we excluded Landsat bands and Landsat-derived spectral indices from the list of predictors used for realized distribution. We then applied our spatio-temporal machine learning framework (feature selection—hyperparameter tuning—ensemble model training) on each species and ran a five–fold spatial cross validation repeated 5 times to evaluate model performances. For the ensemble model we used the same component models (RF, GBT and penalized GLM) and meta-learner (logistic regression). Results of this analysis were then compared with the performances achieved by the ensemble models using Landsat data.
**Table 2 Average logloss for the compared algorithms and for the subset of seven target species.**

| Species | Distribution | ANN | C5.0 | GBT | GLM | KNN | RF | CART |
|---|---|---|---|---|---|---|---|---|
| *A. alba* | Potential | 0.242 ± 0.024 | 0.053 ± 0.009 | 0.097 ± 0.003 | **0.027 ± 0.003** | 0.120 ± 0.021 | 0.033 ± 0.006 | 0.063 ± 0.008 |
| *C. sativa* | Potential | 0.210 ± 0.019 | 0.118 ± 0.020 | 0.128 ± 0.003 | 0.058 ± 0.006 | 0.197 ± 0.033 | **0.057 ± 0.008** | 0.132 ± 0.011 |
| *F. sylvatica* | Potential | 0.516 ± 0.027 | 0.114 ± 0.011 | 0.138 ± 0.003 | **0.055 ± 0.004** | 0.108 ± 0.012 | 0.060 ± 0.004 | 0.184 ± 0.014 |
| *P. abies* | Potential | 0.390 ± 0.017 | 0.176 ± 0.010 | 0.199 ± 0.005 | 0.144 ± 0.006 | 0.314 ± 0.022 | **0.114 ± 0.005** | 0.292 ± 0.009 |
| *P. halepensis* | Potential | 0.220 ± 0.023 | 0.053 ± 0.012 | 0.092 ± 0.003 | **0.019 ± 0.002** | 0.075 ± 0.015 | 0.023 ± 0.003 | 0.070 ± 0.013 |
| *P. sylvestris* | Potential | 0.655 ± 0.041 | 0.370 ± 0.017 | 0.358 ± 0.005 | 0.318 ± 0.008 | 0.569 ± 0.030 | **0.232 ± 0.006** | 0.430 ± 0.013 |
| *Q. robur* | Potential | 0.422 ± 0.019 | 0.117 ± 0.012 | 0.144 ± 0.004 | **0.065 ± 0.004** | 0.152 ± 0.026 | 0.068 ± 0.007 | 0.154 ± 0.010 |
| *A. alba* | Realized | 0.383 ± 0.030 | 0.145 ± 0.017 | 0.140 ± 0.007 | 0.106 ± 0.009 | 0.245 ± 0.059 | **0.059 ± 0.009** | 0.151 ± 0.021 |
| *C. sativa* | Realized | 0.316 ± 0.025 | 0.175 ± 0.031 | 0.161 ± 0.010 | 0.118 ± 0.017 | 0.351 ± 0.065 | **0.077 ± 0.010** | 0.173 ± 0.017 |
| *F. sylvatica* | Realized | 0.654 ± 0.055 | 0.118 ± 0.010 | 0.147 ± 0.005 | 0.129 ± 0.007 | 0.209 ± 0.039 | **0.057 ± 0.011** | 0.200 ± 0.025 |
| *P. abies* | Realized | 0.666 ± 0.053 | 0.180 ± 0.011 | 0.208 ± 0.009 | 0.177 ± 0.009 | 0.467 ± 0.042 | **0.125 ± 0.008** | 0.280 ± 0.035 |
| *P. halepensis* | Realized | 0.292 ± 0.034 | 0.065 ± 0.014 | 0.104 ± 0.004 | **0.025 ± 0.005** | 0.074 ± 0.022 | **0.025 ± 0.005** | 0.084 ± 0.013 |
| *P. sylvestris* | Realized | 0.656 ± 0.043 | 0.451 ± 0.013 | 0.473 ± 0.009 | 0.478 ± 0.009 | 0.776 ± 0.052 | **0.304 ± 0.006** | 0.549 ± 0.012 |
| *Q. robur* | Realized | 0.642 ± 0.049 | 0.105 ± 0.016 | 0.133 ± 0.005 | 0.091 ± 0.011 | 0.200 ± 0.057 | **0.053 ± 0.008** | 0.200 ± 0.030 |

**Note:**
In bold are highlighted the best performing learners for each task.

## RESULTS

### Spatio-temporal machine learning framework

Table 2 shows that RF on average had the highest predictive performances for all species, with GLM coming closer. RF scored the lowest logloss among all the other algorithms in 9 cases out of 14 and scored the same as GLM in 1 case out of 14. In the remaining cases, GLM scored the lowest logloss value, with RF scoring the second lowest. On average, GLM performed better in in potential distribution tasks, with RF clearly outperforming every other algorithm in realized distribution tasks. Overall, GLM and RF always scored the lowest logloss values, from two to three times lower than all the other algorithms in some cases.

The absolute difference between values scored by GLM and RF is lower than when RF had the advantage over GLM. This indicates a high reliability of RF performances even when other models outperform it. The ANN scored the highest logloss values in all tasks, so it was immediately excluded from the pool of level 0 models to choose from. It was time consuming to find a common hyperparameter range well suited for different tasks, since neural networks are often extremely situation-dependent. After a preliminary selection, we used the range shown in Table S3: despite that, our results remained inferior to those obtained with the other learners. On top of that, the *mlr* implementation of neural networks, based on the *deepnet* R package (*Rong & Rong, 2014*), doesn't allow the use of ReLU (rectified linear activation function) as an activation function, which would have been beneficial for our purposes. Based on logloss performances, we selected RF and GLM as the first two components of the ensemble. Based on similar values of logloss (within one standard deviation of the average performance) scored by C5.0, GBT, KNN and CART, we used computational costs to choose the third component model (Table 3).

**Table 3 Hyperparameter tuning, cross validation and prediction time for each model and distribution task.**

| Distribution | Process | ANN | C5.0 | GBT | GLM | KNN | RF | CART |
|---|---|---|---|---|---|---|---|---|
| Potential | Tuning | 561.2 | 310.7 | 527.2 | 448.9 | 2,433.6 | 104.4 | 576.5 |
| Potential | Cross validation | 57.2 | 44.7 | 192.5 | 620.4 | 356.9 | 240.2 | 26.5 |
| Potential | Prediction | 24.1 | 91.4 | 21.9 | 14.8 | 19,272.9 | 35.5 | 15.4 |
| Realized | Tuning | 1,031.6 | 964.2 | 688.4 | 859.1 | 12,321.9 | 396.6 | 1,650.2 |
| Realized | Cross validation | 119.2 | 165.4 | 290.8 | 1,372.9 | 1,445.1 | 805.5 | 114.9 |
| Realized | Prediction | 26.1 | 198.1 | 27.2 | 17.3 | >1 day | 52.4 | 17.4 |
| | Total | 1,819.2 | 1,774.5 | 1,748.0 | 1,455.5 | >1 day | 1,634.6 | 2,400.9 |

**Note:**
Time values are reported in seconds. Tests were conducted in a parallel computing setup on a CPU server running 2 × Intel(R) Xeon(R) Gold 6248R – 3.00 GHz (96 threads) with 504 GB RAM.

KNN was excluded due to computing time values being from one to two order of magnitude higher than the ones scored by the other models. Even though CART scores very low values in cross validation and prediction time in both potential and realized tasks, tuning time is the second highest, just behind KNN. C5.0 is faster than GBT in the whole potential workflow (446.8 s against 741.6) but slower in the realized workflow (1,327.7 s against 1,006.4). Considering both workflows, GBT proved to be faster and more consistent in cross validation and prediction time, showing an increase in tuning time of just 30% with double the amount of training data (see Table S2).

## Variable importance

Of all the features used in both potential and realized distribution, 60 are considered important for both tasks. For potential distribution, diffuse irradiation, precipitation of the driest quarter (BIO17) and precipitation of the driest month (BIO14) were the most important and most frequent predictors across all component models and species (see Fig. 3). The density distributions per macro-class help understanding how the Bioclim macro-class was the one with on average both most important and most frequent variables. Other variables are more species-specific: the minimum surface temperature of April records the highest absolute value in relative importance but it was important for only one species (*Q. robur*, see Fig. S7). The Temperature macro-class accounts the highest numbers of predictors, but the values recorded in both variable importance and frequency are the lowest among all the macro-classes. The Climate macro-class had the largest variety in predictors and variables in this class are homogeneously spread out across all the species in both variable importance and frequency.

For realized distribution, the summer aggregates of Landsat green (25th, 50th and 75th quantiles) were the three most important and most frequent variables across all models and species, closely followed by the summer aggregates of Landsat red and fall aggregates of NDVI (Fig. 3). While the Spectral Index macroclass clearly outperformed the other ones in relative importance, the Bioclim class scored as the most frequent across all the species. The distribution maps scored the highest values for variable importance (distribution of the *F. excelsior* and the *Tilia spp.*) but they were species-specific.

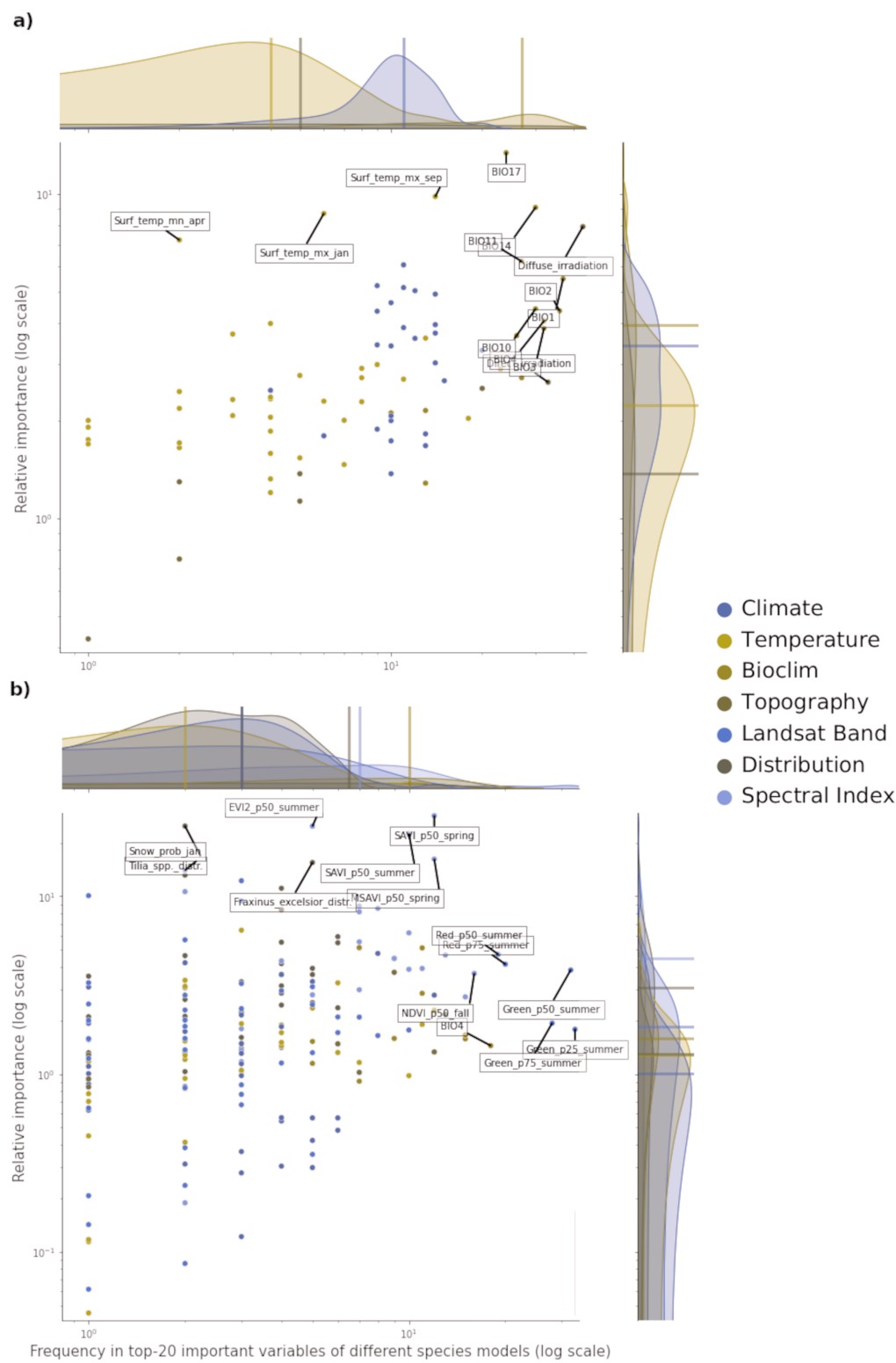

**Figure 3 Relative variable importance *vs* frequency of the variables of the top–20 most important across the component models and all species for potential (A) and realized (B) distribution.** Each plot can be divided in four quadrants, from the top left clockwise: variables with high relative importance

**Figure 3** (continued)
but low frequency (*i.e.*, important for one or few species), variables with high importance and high frequency (*i.e.*, important for all species), variables with low importance and high frequency (*i.e.*, they occured often but were not important) and variables with low importance and low frequency. Labeled dots are variables that recorded high values of relative variable importance or frequency.

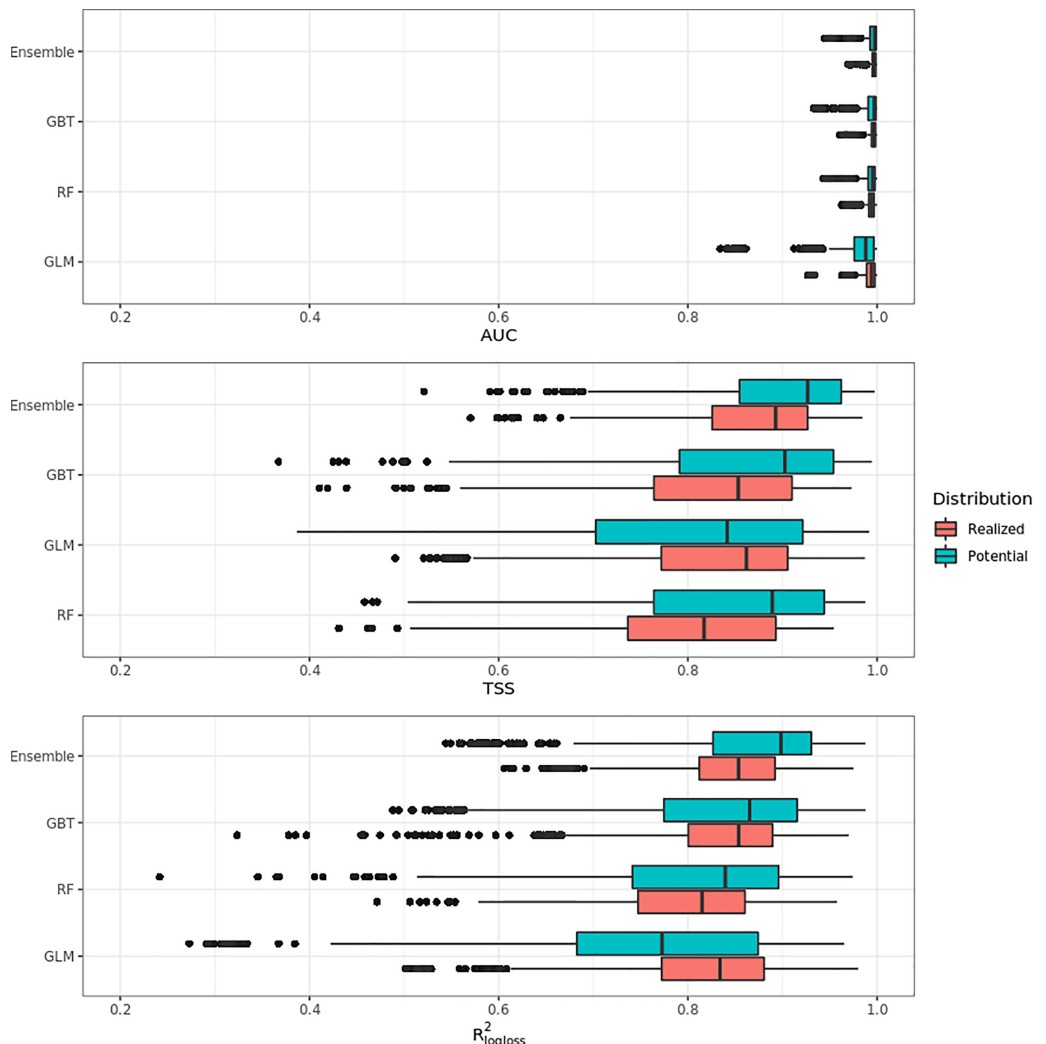

**Figure 4 Aggregated results of the accuracy assessment per model and distribution expressed using AUC, TSS and R²$_{logloss}$.**

Overall, the component models show more differences in variable importance in the potential distribution models than in the realized ones. On average, RF and GBT selected the same variables in the top–10 but not always in the same order, while GLM tended to choose completely different variables (*i.e.*, spectral indices for realized distributions and wind speed for potential distribution). This suggests how the ensemble models tend to use a wider proportion of the feature space than single models. This tendency is most apparent in the potential distribution models. In the realized distribution models, the component

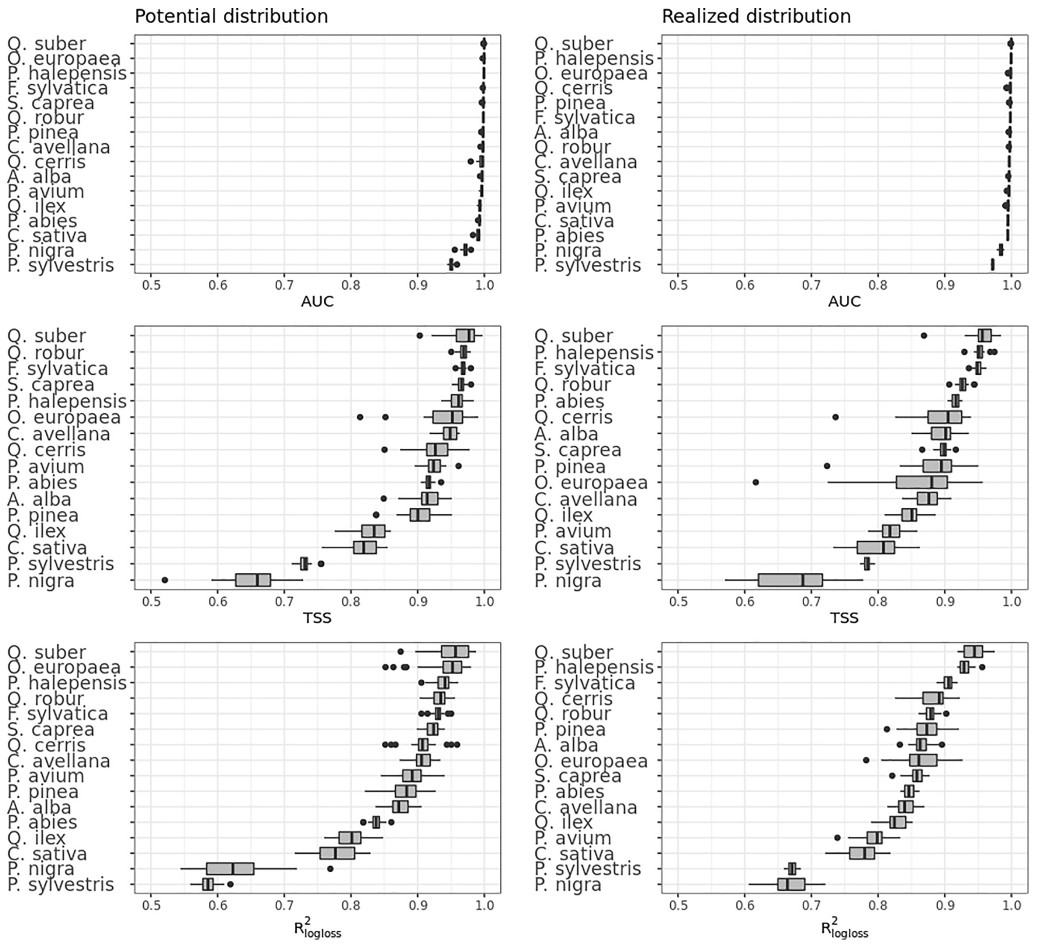

**Figure 5** Results of the accuracy assessment per model and distribution for the ensemble model only expressed using AUC, TSS and $R^2_{logloss}$.

models agree in selecting the top-10 most important variables predictors from Landsat bands or Spectral indices. RF and GBT considered on average the Landsat bands as the most important, while GLM selected the spectral indices more often.

## Accuracy assessment

Figure 4 shows that on average the ensemble model outperformed all component models in both potential and realized distributions. AUC values seem to be overoptimistic and with low variability for all algorithms and distributions, with the largest interquartile range (IQR) being GLM – potential, going from 0.97 to 0.99. Values for TSS and $R^2_{logloss}$ seem to be more conservative, with the ensemble still having the highest average (TSS = 0.898, 0.874 and $R^2_{logloss}$ = 0.857, 0.839, respectively, for potential and realized distribution) values, and lowest IQR (TSS = 0.85–0.96, 0.82–0.92 and $R^2_{logloss}$ = 0.82–0.93, 0.81–0.89, respectively, for potential and realized distribution).

While results from our modeling framework proved GLM and RF being the best models in both potential and realized tasks (see Table 2), GBT achieved overall better performances than both algorithms. In general, the models for potential distribution

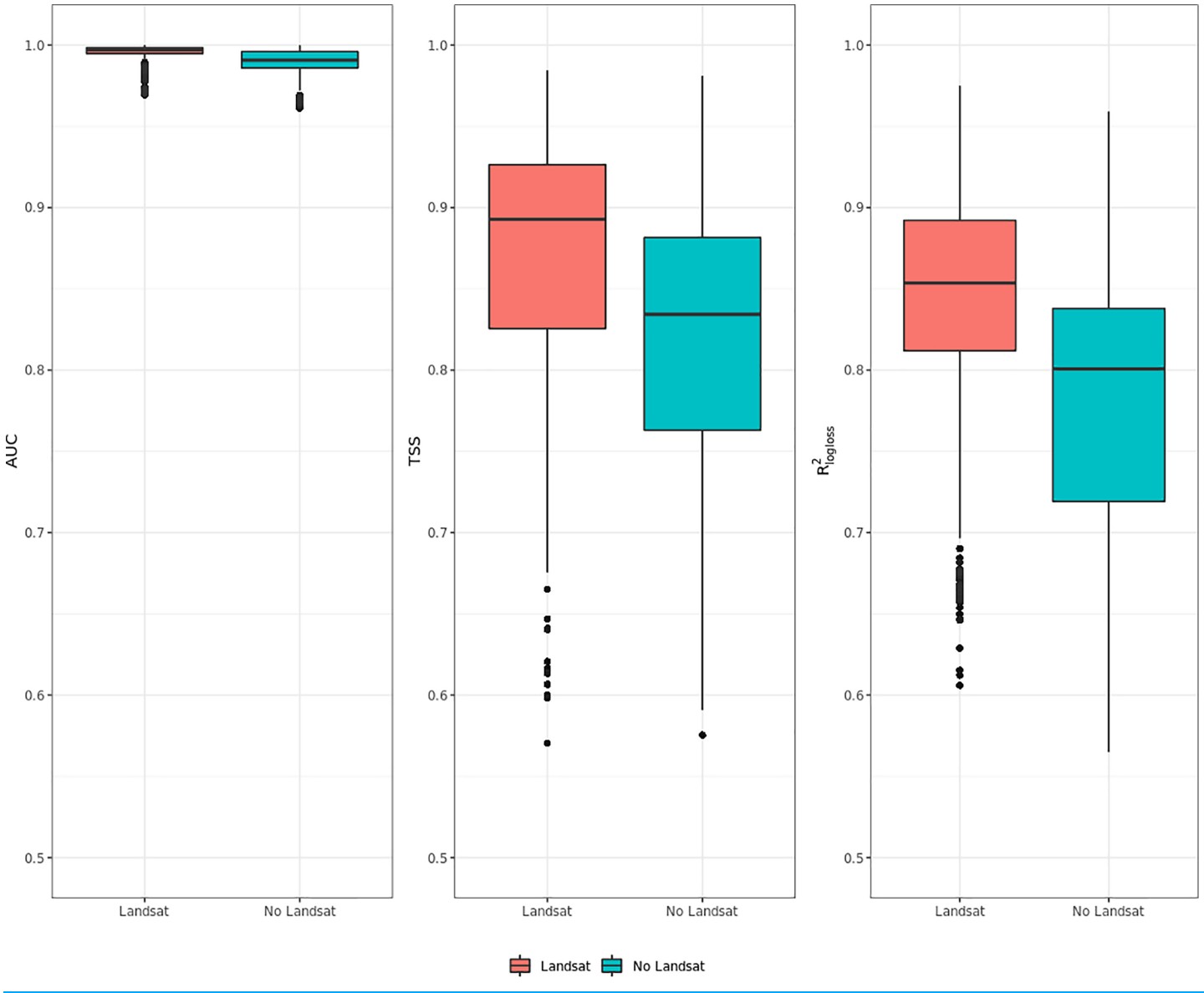

**Figure 6 Aggregated results of the accuracy assessment for modeling realized distribution with and without the Landsat bands and spectral indices expressing using AUC, TSS and $R^2_{logloss}$.**

achieved better predictive performances than those for realized distribution; however potential distribution has greater IQRs as well as larger outliers.

Figure 5 shows the performances of the ensemble model per species and distribution. Results for the component models are available in Table S4 and S5. Following the trend shown in Fig. 4, differences in performances are minimal if we look at the AUC results, while they grow significantly if we look at the TSS and $R^2_{logloss}$ ones. We see that for both potential and realized distribution, models for *Q. suber* achieved the best performances (TSS = 0.968, 0.959 and $R^2_{logloss}$ = 0.952, 0.949, respectively, for potential and realized distribution), while *P. sylvestris* (TSS = 0.731, 0.785 and $R^2_{logloss}$ = 0.585, 0.670) and *P. nigra* (TSS = 0.658, 0.686 and $R^2_{logloss}$ = 0.623, 0.664) achieved the worst.

Furthermore, while *Q. suber* model has overall best performances in all metrics, in both potential and realized distribution AUC grades *P. sylvestris* as the worst and *P. nigra* as the second worst; the opposite is true for TSS scores. For $R^2_{logloss}$, *P. sylvestris* scored as the worst in potential distribution and second worst in realized distribution, with the opposite happening for realized distribution.

### Influence of high resolution on predictive performances

Figure 6 shows that ensemble models for realized distribution including Landsat data consistently outperformed models without Landsat data. In all metrics, values scored by the Landsat models show higher median and average values than the ones without Landsat and lower IQR range. The same trend is shown by all metrics: like in the previous cases shown in "Accuracy assessment", AUC values are high, reaching one for some species, and with low IQR. TSS and $R^2_{logloss}$ show a larger IQR and lower medi an and average values than AUC: given the large differences in values scored by the models in TSS and $R^2_{logloss}$, these two metrics proved to be more helpful to discriminate model performances across multiple species. On average, including Landsat data increases TSS performances by 6.5% and $R^2_{logloss}$ by 7.5%, while when comparing median values the increase in performances is higher in TSS (+7.5%) and lower in $R^2_{logloss}$ (+7%).

## DISCUSSION

### Modeling framework

Combining models using the ensemble approach is thought to reduce model uncertainty and increase its robustness in modeling species distributions (*Araújo & New, 2007*). We used ensemble with stacked generalization as ensemble approach, which has not been tested yet for species distribution modeling. We also trained the models in a spatio-temporal framework, expecting the models to generalize better when predicting in a temporal window not included in the training data.

Part of the intent of the article was to provide a robust reproducible framework to model species distributions based on ensemble ML. *Hao et al. (2020)* used a similar methodological framework to the one used in this study. They modelled the distribution of 13 species of the genus *Eucalyptus* in South Australia and tested performances of ensemble model against individual models; they used *mean* and *weighted average* as ensemble strategies. They also tested cross validation *vs* spatial cross validation for model performances. The study doesn't specify which type of distribution was modelled: according to the definition provided in our study, we can compare their results with our potential distribution results. Their results show how spatial cross validation performances were more conservative than non spatial cross validation ones when compared with performances on independent validation sets. This supports and reinforces our use of spatial cross validation as a validation strategy for the modeling framework. Ensemble models performed well but were outperformed by untuned individual models and by a tuned GBT. There was also no clear advantage in predictive performances when using different ensemble strategies. This is in contrast with our results, where the ensemble based on stacking outperformed even tuned component models (13 cases out of 16), performed
as good as the best component model (2 out 16) and in just one case performed worse (tuned GBT was better than the ensemble). However, this is true only when comparing results from the Logloss and $R^2_{logloss}$: AUC and TSS both show that the ensemble outperformed or performed as good as tuned GBT in all cases and never performed worse. Given the very few occurrences in which the ensemble performed worse, this may be an indication of stacking being a better ensemble strategy when modeling species distribution. *Valavi et al. (2021)* reported an ensemble of tuned individual models as outperforming all other ML and regression based algorithms when benchmarking model performances on potential distribution of 225 different species; their results also show nonparametric technques outperforming traditional regression methods. Among SDM studies focused on testing and comparing different SDM methodologies, the study from *Valavi et al. (2021)* is also one of the few reporting computation time for all the models: this is a metric seldomly reported, but relevant when considering the optimal trade-off between accuracy and time, a well-known issue in the ML field (*Hosseinzadeh et al., 2021*).

In our cross validation estimates, AUC proved the least useful of the performance metrics used in this study, with low variability in AUC scores among different species and distributions despite the difference in predictor variables used and amount and source (*i.e.*, LUCAS, other tree species) of absence data. A general trend shown by the other metrics is mostly picked up (*i.e.*, ensemble superior to base models, models for *Q. suber* being the most accurate), but AUC scores are too similar to ascertain critical problems or possible artifacts in our models. Both TSS and the $R^2_{logloss}$ provided more useful metrics, showing models for *P. sylvestris* and *P. nigra* performing poorly compared to other models in both potential and realized distribution. Our results seem to agree with the ones from *Chakraborty et al. (2021)*, who predicted current and future potential distribution of tree species over Europe using an ensemble framework based on *mean*. We compared our results only with species present in both studies (*A. alba, F. sylvatica, P. abies, P. sylvestris, Q. robur*): final model AUC values are all ≥0.94 on both test set and external validation set, while TSS values start from 0.80.

RF and GLM are the best component models to map both potential and realized distributions when trained on a data sample, but GBT often outperforms RF or even the ensemble when tuned and trained on the whole dataset. In general, differences in predictive performances between the ensemble and the component models are also higher in potential distribution than in realized distribution. The list of variable importance per component model, species and task may give an insight to this: in the potential tasks, the component models use different parts of the feature space before the predictions are combined by the meta-learner. All the models select as most important variables for the task different predictors. For realized distribution tasks, the models all agree in selecting either Landsat bands or spectral indices as most important variables, resulting in predictions that are highly correlated and with less variance between the models.

Ensemble modeling is known to perform best when there is a high diversity between the base models and no or negative correlation between their outputs (*Zhou, 2019*). The introduction of Landsat bands and spectral indices in general greatly increased the predictive performances of the models for realized distribution compared to potential

distribution models. However, this also homogenized predictions, which makes the second condition reported above not always respected. We separately compared the repeated spatial cross validation performances of ensemble and component models excluding the Landsat bands and spectral indices. In this case, the ensemble never performed worse than the best component model. In general, if the ensemble provides predictive performances as good as or worse than the best component model, the best component model must be preferred (*Zhang & Ma, 2012*). However, ensemble models can still provide more advantages than individual models since they reduce model uncertainty and are more robust towards extrapolation (*Mehra et al., 2019*).

High resolution or hyperspectral data have not been used so far for SDMs but are extremely popular in tree species classification: the usage of such predictors has consistently increased over the years the predictive performances of ML tree species classifiers (*Deur, Gašparović & Balenović, 2020*). Such data is not always available on a large spatial and temporal scale, so studies including these predictors usually cover a limited area compared to the one covered by our study. Bridging this gap may help having operational continental scale species distribution maps. Similarly, ML methods have mostly been used for tree species classification, where predictor variables such as temperature or precipitation are seldomly included and environmental variables not throughly considered, rather than for SDM. Despite that, we found in literature several studies which agree with our results: when classifying five (three broadleaves and two conifers) forest tree species in Portugal, *Łoś et al. (2021)* found out that GBT outperformed RF and KNN, reaching accuracy values ≥90% using Sentinel-2 reflectance bands only. *Wessel et al. (2018)* similarly reached high level of accuracy using only Sentinel-2 bands in German forests, but their best results were achieved using an object-based multitemporal Support Vector Machine (SVM) classifier: despite SVM being a very powerful ML method, it was purposefully excluded from this study due to its computation intensity and lack of parallelization.

In our study, the ANN tested performed poorly, mostly due to the limited implementation options in the R environment of this method. Contrary to our results, *Raczko & Zagajewski (2017)* found ANN outperforming RF and SVM when including hyperspectral data; however, they also showed that ANN seemed to be the algorithm which predictions were strictly dependent from the dataset used, while RF and SVM showed more stable performances: the extreme sensibility of ANN to perturbations is a well known issue in the ML field (*Colbrook, Antun & Hansen, 2022*). This issue is partially solved by Convolutional Neural Networks (CNNs), which have achieved considerable results when applied for SDM purposes, even when compared with ML methods such as GBT and RF. CNNs also showed to be particularly promising when commonly used remotely sensed predictor variables such as LiDAR (Light Detection and Ranging) and hyperspectral high resolution data are available (*Zhang, Zhao & Zhang, 2020*; *Fricker et al., 2019*). In some cases, CNNs have even outperformed tuned ML methods given their ability to grasp how local landscape structure affects prediction of species occurrence, in contrast with more conventional ML methods which cannot acknowledge the influence of environmental

structure in local landscapes (*Deneu et al., 2021*; *Sothe et al., 2020*). The ML framework presented in this study could greatly benefit from the inclusion of CNNs.

## Species distributions

Our cross-validation accuracy assessment results indicate high predictive performances for all species, in both potential and realized distributions. In the case of mapping potential distribution, diffuse irradiation and precipitation of the driest quarter (BIO17) are the most important predictors overall. These results are partially in contrast with *Dyderski et al. (2018)*, who modelled current and future potential distribution of 12 tree species over Europe. We compared our results only with species present in both studies (*A. alba, F. sylvatica, P. abies, P. sylvestris, Q. robur*): in their case, temperature-related bioclimatic variables (BIO1, BIO5, BIO7 and BIO10) were more important than precipitation-related bioclimatic variables. Few peer-reviewed studies have reported on the importance of predictors other than bioclimatic ones in shaping species' potential distributions.
We found that, on average, each component model considers two or more predictors from the Bioclim macro-class among the top-10 most important variables to predict the potential distribution. Previous findings in literature have shown the importance of bioclimatic variables when modeling species distributions (*Fourcade, Besnard & Secondi, 2018*), but this may also be a consequence of bioclimatic variables and elevation being the most employed, if not the only, predictors in numerous SDM studies (*Fois et al., 2018*). *Bucklin et al. (2015)* compared the influence of different sets of environmental predictors on model performances, but the list of predictors used in the study included human influenced factors, so their results cannot be used to assess the driving factors for potential distributions. Even if our results show the bioclimatic variables as the most important predictors for potential distributions, further studies in this direction may be needed.
The scale of the study may affect the importance of predictor variables: on a large scale, distribution may be influenced by macro environmental factors, while at a local scale, other environmental factors may limit distribution more significantly. *Walthert & Meier (2017)* and *Weigel et al. (2019)* proved that soil properties are more important than either bioclimatic or only climatic variables when modeling potential tree species distribution at, respectively, country and regional scale.

Variable importance confirms that Earth Observation layers such as the 25th, 50th and 75th quantile summer aggregates for the Landsat green and red band and the 50th quantile fall aggregates of NDVI are overall the most important layers for mapping realized distribution of species. The inclusion of Landsat data and derived spectral indices increases predictive performances and contains more detailed information on species distribution ranges. Importance of NDVI is well known since it is one of the most used proxies in vegetation studies such as biodiversity estimation (*Madonsela et al., 2017*; *He, Zhang & Zhang, 2009*), net primary productivity (*Schloss et al., 1999*) and land degradation (*Easdale et al., 2018*), phenology (*Fawcett, Bennie & Anderson, 2021*) and species composition changes (*Wang et al., 2021*). NDVI incorporates information from the red and the

near-infrared (NIR) portion of the electromagnetic spectrum. Vegetation's behavior in this portion of the spectrum has long been used in vegetation mapping to distinguish between coniferous and deciduous tree species (*Hoffer, 1984*). The green band, although usually less important than the red and NIR band, has already proved useful in vegetation mapping to classify forest types (*Gao et al., 2015*), predict forest variables (stem volume, diameter and tree height) at species level (*Astola et al., 2019*) and forest biomass at community level (*Nandy et al., 2017*).

Comparing our results with chorological maps from the European Atlas of Forest Tree Species (*San-Miguel-Ayanz et al., 2016*), we can see that in general both potential and realized distribution correctly capture the species ranges. Overall, potential distribution maps show homogeneous patterns of high probability values for all target species, while realized distribution maps show very fragmented patterns. The realized distribution model helps discriminating the presence or absence of the species due to biotic or other external factors. A high geographical overlap between probability maps of realized distribution of different species may reflect co-existence within the same forest stands and could help in clearly define forest communities.

However, our results have to be interpreted and analyzed carefully: contrary to process based models, correlative models describe the patterns, not the mechanisms, in the association between species occurrences and predictor variables; for this reason, correlative SDMs risk overlooking potentially important driving factors determining species distributions since they cannot distinguish between direct and indirect effects (*Sirén et al., 2022*). A known issue in this sense is the masking effect of abiotic factors on competition and predation: SDMs could estimate abiotic predictors as the most important for species abundance, even in those cases when distribution is strongly affected by competition (*Godsoe, Franklin & Blanchet, 2017*) or when biotic interactions are strictly correlated with abiotic factors (*Filazzola, Matter & Roland, 2020*).

ML methods strongly depend from high quality datasets, so a considerable effort was spent in creating two different presence-absence datasets for each target species, one for potential and one for realized distribution: while the same geographical extent was used for both datasets, different rules were used to select true absence (LUCAS dataset) or pseudo-absence (other tree species occurrences) points. This modeling choice may be one of the causes of cross validation estimates for potential distribution being higher than for realized distribution. Restricting the study area from which true and pseudo absence points are collected reduces the applicability of the models for predictive purposes (*Pearson & Dawson, 2003*), with unpredictable effects on future projections (over- or under-prediction, see *Thuiller et al. (2004)*). On the other end, no spatial constraint leads to unwanted situations where larger scale differences rather than local ones are picked, leading to the infamous SDM case of "there-are-no-polar-bears-in-the-Sahara" (*Lobo, Jiménez-Valverde & Hortal, 2010*). *Chefaoui & Lobo (2008)* proved how best hypotheses for potential distribution are obtained using absence points that are placed farther apart than the ones needed for the best hypotheses on realized distribution, hence using the same

study area for both distributions may have lead to an overstimation of presence in potential distribution. Furthermore, *Lobo, Jiménez-Valverde & Hortal (2010)* proved that for realized distribution best practice would be to avoid the absences from nearest localities due to possible contamination with methodological absences; while this was not an issue when considering absences coming from the LUCAS dataset, using information of other tree species occurrences as pseudo-absence may have affected our models. The criteria used to select pseudo-absence occurrences in SDM represent a big challenge in SDM, such that the topic has been the focus of multiple studies in the last decade (*Iturbide, Bedia & Gutiérrez, 2018a*, *2018b*; *Senay, Worner & Ikeda, 2013*).

## High resolution contributions: is finer always better?

Bioclimatic variables available only at coarse spatial resolution were used as predictor variables in both potential and realized distribution. The Landsat bands and the spectral indices were not the only high resolution layers used in this study: terrain and terrain-derived predictors were also included at 30 m resolution. However, despite the terrain data high resolution, the tree species potential distribution patterns mostly reflect the original spatial resolution of the bioclimatic variables. Thus, climate influences species distribution at the European scale. Even though this might indicate that mapping potential distributions at high resolution may not be necessary, it can still be useful for different case studies. For example, comparing the difference, and hence mapping the gap, between potential and realized distribution at the same fine scale may prove to be an invaluable tool for both forest managers and conservation planners that work on the local level.

Potential distribution maps can be used to identify suitable areas for species in reforestation and restoration programs; realized distribution maps can inform the forest managers on the presence or absence of said species in those areas at a particular point in space and time (Fig. 7). By removing the biotic factors that limit the presence of the species in a potential reforestation site, using multiple distribution maps and including expert knowledge on species synecology, structurally complex forest stands could be planned and developed in a much more informed and data-driven way. A similar approach could be used by conservation planners. Potential distribution is modelled by studying the relationship between a species and the environmental conditions found in its native range, where the species is at equilibrium (*Jiménez-Valverde et al., 2011*). Invasive species are usually more abundant and productive in the introduced range than in their native ranges (*Hierro, Maron & Callaway, 2005*). This is due to the absence of biotic factors that normally limit species distribution in their native range in the introduced range. Thus, a species that occupies only 10% of its potential distribution in its native range may end up occupying a bigger percentage of it in the introduced range. Estimation of potential distribution in the introduced range that depends only on environmental factors are conservative by definition, potential distribution maps may provide a good indication to conservation planners of how much the invasive species could spread in the introduced range.

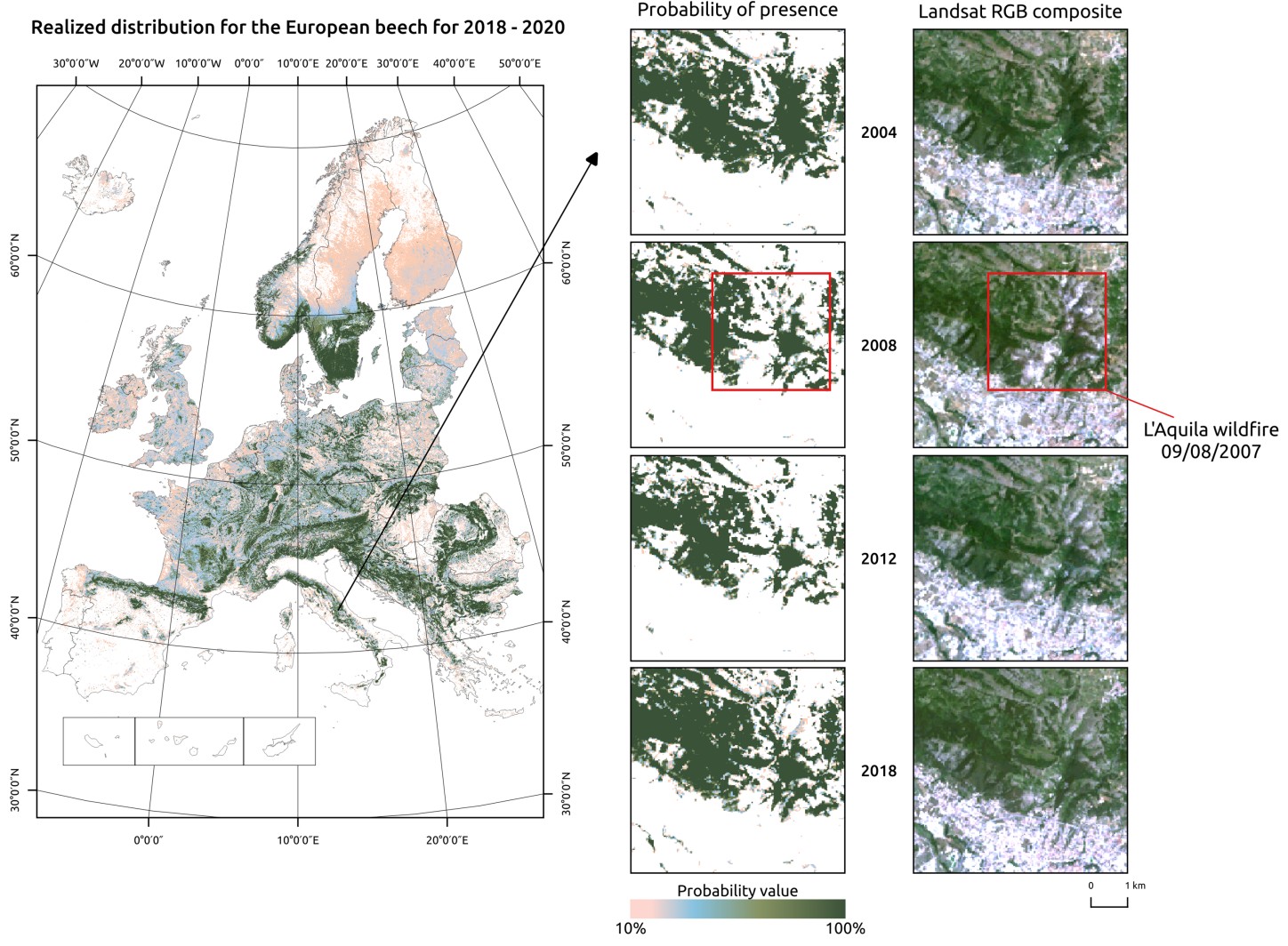

**Figure 7 Realized distribution of *Fagus sylvatica* for the period 2018–2020.** Detailed insets show a region around L'Aquila city, in Central Italy. The *Fagus sylvatica* forest on the northern outskirts of the city was affected by a serious wildfire in 2007. The realized distribution maps can be used to track compositional changes through time.

For realized distribution, including high resolution predictor variables in the model not only increases predictive performances but also lowers overall and local values of uncertainty. For forest management purposes, a large, consistent, standardized, long-term and high resolution image collection such as the one provided by the Landsat program can help extending in space and time information on tree species presence, composition and abundance. A spatial resolution of 30 m is particularly well suited for NFI applications: *Strickland et al. (2020)* derived probability maps of forest tree species for a 25 years time period (1985–2010) using yearly Landsat composites to extend missing information from the Canadian NFI and estimating changes in forest cover, species composition and forest disturbances. The increasing availability of even higher-spatial resolution satellite data from the European Copernicus program (*i.e.*, Sentinel 1 and 2) and commercial providers

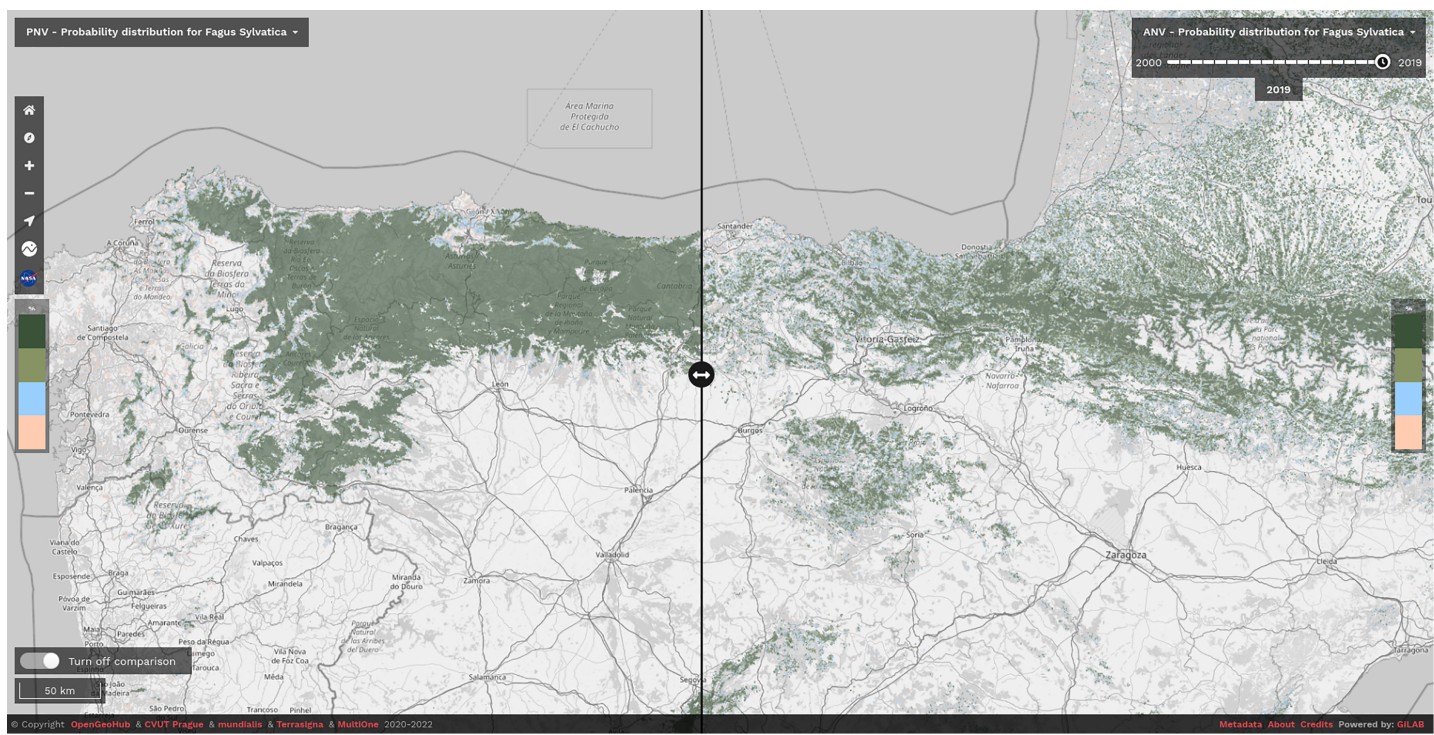

**Figure 8** **Difference between potential and realized distribution for** *Fagus sylvatica* **in Northern Spain for the period 2018–2020 visualized using slider in the Open Environmental Data Cube Europe viewer (https://ecodatacube.eu).** ©Copyright OpenGeoHub & CVUT Prague & mundialis & Terrasigna & MultiOne 2020–2022.

(*i.e.*, Planet) can potentially further enhance predictions by including more data and a better spatial matching of *in-situ* and satellite-derived information.

## CONCLUSION

In this article we have developed, tested and reported a methodological framework for predicting potential and realized distributions of 16 forest tree species using ensemble ML and analysis-ready EO data. In general, our ensemble models achieved better predictive performances than individual models when modeling both potential and realized distribution, while performing as good as the best individual model in the worst cases. Bioclimatic variables proved in general to be the most important and frequent predictors for potential distribution across Europe, mainly through precipitation-related predictors (BIO17 and BIO14) even at high resolution (*i.e.*, on a local scale), while reflectance-based covariates were the most important predictors of the realized distributions. Overall, realized distribution proved to be more complex to map accurately than potential distribution and, among the species analyzed, distributions of specialist species proved easier to classify than pioneer species. In general, the ensemble and component models achieved better predictive performances for the potential distributions than for the realized distributions as judged from the cross-validation estimates. Our results indicate a consistent increase in predictive performances for realized distribution when adding high resolution data, especially Landsat data at 30 m resolution and spectral indices to the list of

predictors. Significant findings of our work include: (**a**) distribution mapping for forest tree species can be efficiently automated to the level of full automation, but this assumes high quality/artifact free training points with a homogenous distribution of occurrence and absence points whenever possible; (**b**) complexity of ML methods can be significantly reduced by implementing efficient hyperparameter tuning and feature selection; (**c**) analysis-ready, high resolution reflectance time-series layers are maybe cumbersome to prepare and gap-fill for clouds and artifacts, but overall come as the most important inputs for maximizing predictive performances of realized tree species distribution.

We have released the maps and the code under open data/open source licenses to enable other similar research and to help speed up land restoration and reforestation projects in Europe. The code is publicly available in our GitLab repository at https://gitlab.com/ geoharmonizer_inea/spatial-layers/-/tree/master/veg_tree.species_anv.pnv.eml, while the datasets and predictions of tree species are available as Cloud-Optimized GeoTIFFs on Zenodo (see https://doi.org/10.5281/zenodo.5818021) and can be displayed in 2D and 3D using the compare tool on the Open Environmental Data Cube Europe viewer (see Fig. 8).

Even though we achieved high values of predictive performances, we still recognize many future areas of improvements. Given the importance of Landsat data for the results of this study, using a larger and higher resolution stack of reflectance-based predictors could help to improve precision of the predictions. A good example in this direction would be fusing all EO data currently available such as Harmonized Landsat Sentinel-2 (HLS) (*Claverie et al., 2018*) and eventually Sentinel 1 datasets. Hyper-spectral images (*i.e.*, from future hyper-spectral missions such as ENMAP, see https://www.enmap.org/) are also proving to be useful to discriminate between different tree species and especially those that grow under dominant species (*Fricker et al., 2019*; *Shen & Cao, 2017*). As any ML-derived product, our predictions would benefit from having more and better quality data on tree species, in particular those that come from NFI plots: it is now crucial to have such data freely available to monitor processes such as species compositional changes, niche shifts, forest regrowth and degradation, as recently stated by *Nabuurs et al. (2022)*. Exploring more sophisticated and different ML algorithms such as Deep Learning (DL) techniques (*Lakshminarayanan, Pritzel & Blundell, 2016*) for our ensemble framework is also another area of improvement given the wide variety of applications these methods possess and the results obtained in comparison with other conventional ML algorithms (*Choe, Chi & Thorne, 2021*; *Deneu et al., 2021*; *Anand et al., 2021*).

European forest dynamics, even though some recent results indicate increased mortality in European forests (*Popkin, 2021*; *Senf, Sebald & Seidl, 2021*; *Senf et al., 2018*), are probably among the least troubling in comparison to other continents. Our methodological framework could potentially be implemented at a global scale, and possibly through Google Earth Engine (GEE) (*van den Hoogen et al., 2021*) or through the European Space Agency's OpenEO platform (https://openeo.cloud/) to produce high resolution (10–30 m) predictions of forest dynamics. Globally, there are many more tree species which are more important for forest management and monitoring. For example, South America as a whole has 4 times the amount of tree species present in Europe and 50% of all tree species on Earth (*Cazzolla Gatti et al., 2022*); in Brazil, it has been estimated

that about 220 tree species cover most of the land and represent over 95% of the biomass (*i.e.*, so called *"hyper-dominant species"* (*Draper et al., 2021*)). Scaling up the approach described in this article to help producing objective predictions, to assist with monitoring forest dynamics and to support reforestation efforts globally is part of our next objectives.

## ACKNOWLEDGEMENTS

We are grateful to the GiLAB company from Belgrade, Serbia for their support with processing and publishing produced data *via* the opendatascience.eu data portal. We are also grateful to the Geo-harmonizer project partners CVUT Prague, mundialis, Terrasigna & MultiOne for helping with processing all LUCAS ground observations and quality control.

### Funding

This work is financed under Grant Agreement Connecting Europe Facility (CEF) Telecom project 2018-EU-IA-0095 by the European Union (https://ec.europa.eu/inea/en/connecting-europe-facility/cef-telecom/2018-eu-ia-0095). There was no additional external funding received for this study. The funders had no role in study design, data collection and analysis, decision to publish, or preparation of the manuscript.

### Grant Disclosures

The following grant information was disclosed by the authors:
Grant Agreement Connecting Europe Facility (CEF) Telecom Project: 2018-EU-IA-0095.

### Competing Interests

The authors declare that they have no competing interests.

Carmelo Bonannella, Tomislav Hengl and Leandro Parente declare that they are officially employed by OpenGeoHub.

### Author Contributions

- Carmelo Bonannella conceived and designed the experiments, performed the experiments, analyzed the data, prepared figures and/or tables, authored or reviewed drafts of the article, and approved the final draft.
- Tomislav Hengl conceived and designed the experiments, authored or reviewed drafts of the article, and approved the final draft.
- Johannes Heisig performed the experiments, analyzed the data, authored or reviewed drafts of the article, and approved the final draft.
- Leandro Parente conceived and designed the experiments, performed the experiments, authored or reviewed drafts of the article, and approved the final draft.
- Marvin N. Wright conceived and designed the experiments, authored or reviewed drafts of the article, and approved the final draft.
- Martin Herold conceived and designed the experiments, authored or reviewed drafts of the article, and approved the final draft.

- Sytze de Bruin conceived and designed the experiments, authored or reviewed drafts of the article, and approved the final draft.

## Data Availability

The input data is available at Zenodo: Bonannella, Carmelo, Hengl, Tomislav, Heisig, Johannes, Leal Parente, Leandro, Wright, Marvin, Herold, Martin, & de Bruin, Sytze. (2022). Presence-Absence Points for Tree Species Distribution Modelling for Europe (0.3) [Data set]. Zenodo. https://doi.org/10.5281/zenodo.6516590.

Outputs are split in different Zenodo entries, they are referenced as associated references in the Zenodo repository with the input data.

The predictions are also available at Open Data Science: https://maps.opendatascience.eu/.

The code is available at GitLab: https://gitlab.com/geoharmonizer_inea/spatial-layers/-/tree/master/veg_tree.species_anv.pnv.eml.

The tutorial is available at Spatiotemporal interpolation using Ensemble M: https://opengeohub.github.io/spatial-prediction-eml/spatiotemporal-interpolation-using-ensemble-ml.html#spatiotemporal-distribution-of-fagus-sylvatica.

Additional material is available on Zenodo: Bonannella, Carmelo, Hengl, Tomislav, Heisig, Johannes, Leal Parente, Leandro, Wright, Marvin, Herold, Martin, & de Bruin, Sytze. (2022). Supplemental material for "Forest tree species distribution for Europe 2000–2020: mapping potential and realized distributions using spatiotemporal Machine Learning" (0.1). Zenodo. https://doi.org/10.5281/zenodo.6516728.

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
