# Peer review of "Forest tree species distribution for Europe 2000–2020: mapping potential and realized distributions using spatiotemporal machine learning"

_PeerJ, doi:10.7717/peerj.13728_

## Round 0.1 · original submission · Major Revisions

I have now received two independent reviews on your manuscript. While both saw merit in it, they also pointed out many aspects that demand the author's attention. I'd like to highlight R1's comment that the manuscript looks more like a student thesis. It's too long and the writing style needs to be reformulated to be more appropriate as a journal paper. R2 makes many good suggestions in that regard. The Methods section is gigantic and needs to be shortened very much. More other technical details in the supplementary material. The Results need to be better summarized by only reporting the key findings in the main text. You have a lot of predictor variables. How did you deal with multicollinearity? The background theory on SDMs should be deleted; just point readers to key papers instead. The introduction needs to actually point the reader to knowledge gaps and inform him/her of the importance of the study. Now it reads more like a literature review.

Like R1, I don't think you need to report results of the potential distribution at all. So, unless you have a good reason to keep them in the manuscript, remove them. Additionally, double-check the way you dealt with absences. Both reviewers pointed out some problems with that. R2 also comments on a potential problem with spatial thinning. I recommend authors to take a closer look at the spThin R package:

https://onlinelibrary.wiley.com/doi/abs/10.1111/ecog.01132

Also, I'd kindly ask authors to adhere to the two standards for reporting and securing reproducibility of SDMs:

https://onlinelibrary.wiley.com/doi/full/10.1111/ecog.04960

https://www.nature.com/articles/s41559-019-0972-5

Please, provide the documents suggested by these authors as an appendix.

I have made several comments in the pdf attached. Please, also refer to it when preparing the revised version of the manuscript.

·

Basic reporting

This study is useful and innovative in combining remotely sensed and other types of variables for tree species mapping, and in developing temporally dynamic maps (for 4 year time steps). The temporal mapping is a very nice contribution of this study.

I am not really sure how useful it is to use realized distributions to evaluate the predicted potential distribution. I mean, it is not surprising that the accuracy is lower.

I think the background section on SDM theory lines 158-218 could be greatly condensed to highlight just the two relevant points – potential versus realized (although as noted above, I am not sure the ‘potential’ modeling is a very informative part of this paper) and the issue of presence/absence data (a great strength of your study is that you have p/a data, and many do not).

I think you missed this opportunity because the paper is couched too much in the framework of SDM (high spatial resolution data have been underrepresented as predictors in SDM). I think this is because tree species mapping studies have been represented more in the remote sensing image classification literature – where one could say that non-image predictors are underrepresented! Terrain and some other non-image data have been incorporated into image classification, however, since the 1980s.

Experimental design

I am surprised that you did not test convolutional neural networks because they seem to be promising for tree species identification from hyperspectal high resolution and lidar data. Perhaps other ML approaches are something you could discuss in your paper even if you did not test them.

Fricker, G. A., Ventura, J. D., Wolf, J. A., North, M. P., Davis, F. W., & Franklin, J. (2019). A convolutional neural network classifier identifies tree species in mixed-conifer forest from hyperspectral imagery. Remote Sensing, 11(19), 2326.

I also feel that your paper needs more extensive comparison with other studies using ML to classify forest tree species (especially in Europe). That would really help put your results in context. This one looks relevant:

Raczko, Edwin, and Bogdan Zagajewski. "Comparison of support vector machine, random forest and neural network classifiers for tree species classification on airborne hyperspectral APEX images." European Journal of Remote Sensing 50, no. 1 (2017): 144-154.

Validity of the findings

I do not really like your use of non-forest land cover as absences – you should really use locations of other species (other than the target species) if you are trying to discriminate tree species.

I appreciate the reporting of hyperparameters.

Logloss might be robust but nobody uses it to report accuracy in SDMs. In order for your results to be more useful and the paper to be read and cited you should report a more widely used accuracy measure.

Reviewer 2 ·

Basic reporting

see additional comments

Experimental design

within scope; relevant interesting study; see additional comments on methods

Validity of the findings

see comments

Additional comments

Review of Peerj 69672-v0
Overview
This manuscript reports on an effort to map the actual and potential distributions of 16 European tree species. A single potential distribution map for each species was generated for the 2000-2020 period, while maps of actual distribution were generated for every 4th year, resulting in a 6-map time series over this period. The study employs an impressive state-of-the-art modelling approach that features: 3 million occurrence locations, six machine learning algorithms, a stack of >500 explanatory variables (including climate, soil, spectral reflectance, and biotic variables), and generation of ensemble models using a meta-learner for each species. The resulting maps provide high-resolution (30 m) occurrence probabilities for each species across the European study area, with the following key findings: high-resolution inputs and outputs significantly improved map accuracy, predictor variable importance varied across species and models, and ensemble models tended to outperform individual models. This study employs high technical standards to address an important topic for a variety of ecology-related fields. Below we make a number of suggestions to improve the manuscript.
Main Comments
1. The authors purport to model species’ potential distributions by: 1) carefully selecting absence points for each species (i.e., only points falling outside both human-altered land cover types and existing tree atlas range boundaries), and 2) including only certain environmental predictor variables in the modelling process (i.e., no biotic variables or Landsat imagery). At line 182, they define the potential niche as “that portion of the fundamental niche that exists in the study area at the time of the study (Peterson et al., 2011)” – in other words, anywhere the species could become established in the study area. If this is truly what they are attempting to model, then they are surely not accomplishing it by controlling absence locations and predictor variables. As the authors point out at lines 178-181, true potential distribution cannot be modelled using a correlative approach; this is more the realm of mechanistic models. Thus, it would seem that the products produced here are a hybrid between actual and potential distributions. The authors should further clarify the exact nature of their potential distribution maps.
2. Multiple studies have suggested that the study area from which absence or background points are selected should be carefully considered as it has important implications for model outputs. For the current work, the study limits include the entire European continent, but there is no discussion of the implications this may have on the resulting models.
3. The spatial distribution of presence/absence points is an important aspect of species distribution modelling. In the current study, the very short description of the ‘thinning operation’ that was applied to occurrence locations (lines 444-448) is simply not detailed enough to allow readers to assess this key aspect of the study.
4. In Section 4.3 of the manuscript, there is considerable discussion around the relative importance of the various explanatory variables in determining species distributions. A well-known challenge of such efforts is that correlations between explanatory variables make it difficult to attribute causation to any particular variable(s). The authors should at least provide a caveat that recognizes this inherent limitation of correlative modelling approaches.
5. The quality of the writing throughout the manuscript is generally fine, but there are a large of grammatical improvements that could be made. I have suggested a number below, but further efforts in this area would be beneficial. Perhaps an English editor could have a close look? Even the last sentence in the manuscript as written is awkward/not grammatically correct and requires an edit: Scaling up approach described in this paper to help produce objective predictions and help monitor forest dynamics and support re-forestation efforts across globe is our next frontier.
6. Stylistically, the number of lists could be greatly reduced, which could potentially shorten the manuscript by several pages. Examples include: 1) bullets of time periods at lines 236-242; 2) numbered list of 16 species at lines 258-274; 3) bullets of forest types at lines 298-304; 4) bullets of land cover types at lines 314-322; 5) bullets of ML algorithms at lines 482-490; see also lines starting at 574. These are just a few examples. The large number of lists and bullets throughout the manuscript take up unnecessary space and, in my opinion, make it feel more like a report than a scientific article.
Specific Comments
Line 19; replace ‘Paper…’ with ‘This paper…’ .
Line 40; acronyms NDWI and NDVI have not yet been defined.
Line 44; acronym EO has not yet been defined.
Line 49; replace ‘restauration’ with ‘restoration’.
Line 54-56; this sentence seems to be a bit of an overstatement – if we could ‘anticipate and minimize climate change impacts’ simply by knowing where tree species occur, I’d be a lot more optimistic about the future of our forests!
Line 73; insert the word ‘an’ between ‘ML’ and ‘increasingly’.
Line 74; replace the word ‘an’ with ‘a large’.
Line 78; replace ‘presence-absence we find’ with ‘for presence-absence modelling are’.
Line 100; replace ‘effect’ with ‘effects’.
Line 109; never heard of ‘auto-ecology’… do you mean autecology?
Line 114; insert the word ‘are’ between ‘that’ and ‘potentially’.
Line 116; replace ‘Earth Observation’ with ‘EO’ as you have already defined it at Line 93.
Line 121; insert the word ‘that’ between ‘fact’ and ‘the’.
Line 156; replace ‘whole of’ with ‘the entire’.
Line 177; the acronym ‘BAM’ should be defined – e.g., Biotic-Abiotic-Mobility (BAM) diagram.
Line 195; replace ‘vegetation’ with ‘plant’.
Line 195; replace ‘owing to’ with ‘by’.
Line 197; replace ‘just temporary’ with ‘temporarily’.
Line 198; remove the word ‘of’.
Line 311; replace ‘was’ with ‘were’.
Line 312; remove sentence starting ‘As for the occurrence points…’ as it is redundant with sentence starting with ‘Each unique combination…’ at Line 306.
Line 340; the acronym ‘GLAD’ has not been defined.
Line 344; replace ‘period’ with ‘periods’.
Table 1, column 1, row 3; replace ‘Soild’ with ‘Soil’.
Line 353; replace ‘have’ with ‘generate’.
Line 459; replace ‘metric’ with ‘metrics’.
Line 495; replace ‘an hyperparameter’ with ‘a hyperparameter’.
Line 504; replace ‘educes’ with ‘reduces’.
Line 528; replace ‘towards’ with ‘regarding’.
Line 537; replace ‘performances’ with ‘performance’.
Line 574; consider removing the raw R outputs associated with the final fitted ensemble model. Is it really necessary? Could it be presented in a table or in the supplementary material?
Line 697; replace ‘underestimations’ with ‘underestimation’.
Line 699; correct the term ‘datacube cube’.
Line 731; consider replacing ‘not tuned’ with ‘untuned’.
Line 768; replace ‘come as’ with ‘are the’.
Line 771-774; replace:
‘Which environmental variables and their relative importance as limiting factors are still unclear for many tree species and few peer reviewed studies focused on investigating the importance of different environmental predictors on potential distributions are available.’
With:
‘Few peer-reviewed studies have reported on the importance of various environmental predictors in shaping species' potential distributions.’
Line 788; not clear why mangrove distributions are being referenced here?
Line 791; remove ‘For realized distribution…’ from the start of the sentence as it is repeated at the end of the sentence.
Line 820; replace ‘…regardless of the terrain data high resolution,…’ with ‘…despite the high resolution of the terrain data,…’
Line 822; replace ‘on’ with ‘at the’.
Line 832; please clarify the meaning of: ‘The opposite approach could be used by conservation planners’.
Line 835; replace ‘have greater performances’ with ‘productive’.

---

## Round 0.2 · Minor Revisions

I have now received one assessment of your revised manuscript by one of the former reviewers. Like her, I believe the text has improved considerably and now looks more like an actual research paper. However, the introduction still needs further improvement to make the ideas and connections between paragraphs better. The conclusion is still larger than what would be ideal, try to reduce it to a couple of key messages only. It currently has 10 paragraphs, try to condense it to a maximum of three paragraphs.

·

Basic reporting

The authors have addressed all of the substantive criticisms to my satisfaction. I think overall that the revised manuscript is now suitable for publication. This is a valuable and exciting study and an impressive amount of data processing.

The introduction, although the content and focus is much improved, is a bit choppy. All the right parts are there but the connections between the parts do not flow. For example the 2nd paragraph, line 57-, is not really a paragraph and should be integrated with the 3rd. The paragraphs jump from 1. Importance of forests and mapping them, 2. What is SDM, 3. Spatio-temporal and EO data have not been used much in SDM (these are both important points, and maybe deserve more detail – much temporal remote sensing is used for phenology and not for high-resolution mapping), 4. Ensemble and ML modeling. It needs smoother transitions or connectors. I really recommend the following book for helping scientific writers follow s story arc, introduce the main characters, etc.
Schimel, J., 2012. Writing science: how to write papers that get cited and proposals that get funded. OUP USA.


The manuscript may require a careful final editing to catch all remaining minor grammatical errors and typographical errors.

Experimental design

Good

Validity of the findings

Good

Additional comments

none

---

## Round 0.3 · accepted · Accept

Thank you for making those final amendments and corrections to the manuscript. I believe it's now ready to be published and I anticipate it'll make a great impact on the field.